# Predictive immunoinformatics reveal promising safety and anti-onchocerciasis protective immune response profiles to vaccine candidates (*Ov*-RAL-2 and *Ov*-103) in anticipation of phase I clinical trials

**Derrick Neba Nebangwa**[1◉]*, **Robert Adamu Shey**[1,2◉], **Daniel Madulu Shadrack**[3], **Cabirou Mounchili Shintouo**[1], **Ntang Emmaculate Yaah**[1], **Bernis Neneyoh Yengo**[4], **Mary Teke Efeti**[1], **Ketura Yaje Gwei**[1], **Darling Bih Aubierge Fomekong**[5], **Gordon Takop Nchanji**[2,6], **Arnaud Azonpi Lemoge**[7], **Fidele Ntie-Kang**[8,9,10]*, **Stephen Mbigha Ghogomu**[1]*

1 Department of Biochemistry and Molecular Biology, Faculty of Science, University of Buea, Buea, Cameroon, 2 Tropical Disease Interventions, Diagnostics, Vaccines and Therapeutics (TroDDIVaT) Initiative, Buea, Cameroon, 3 Department of Chemistry, St. John's University of Tanzania, Dodoma, Tanzania, 4 Department of Microbiology and Immunology, College of Medicine, Drexel University, Philadelphia, Pennsylvania, United States of America, 5 Department of Nursing, Mount Zion Higher Institute, Mankon, Bamenda, Cameroon, 6 Department of Microbiology and Parasitology, Faculty of Science, University of Buea, Buea, Cameroon, 7 Ngonpong Therapeutics, Concord Pike, Wilmington, Delaware, United States of America, 8 Center for Drug Discovery, University of Buea, Buea, Cameroon, 9 Department of Chemistry, University of Buea, Buea, Cameroon, 10 Institute of Pharmacy, Martin-Luther University of Halle-Wittenberg, Halle, Germany

◉ These authors contributed equally to this work.
* neba.nebangwa@ubuea.cm (DNN); stephen.ghogomu@ubuea.cm (SMG); fidele.ntie-kang@ubuea.cm (FNK)

## Abstract

Onchocerciasis (river blindness) is a debilitating tropical disease that causes significant eye and skin damage, afflicting millions worldwide. As global efforts shift from disease management to elimination, vaccines have become crucial supplementary tools. The Onchocerciasis Vaccine for Africa (TOVA) Initiative was established in 2015, to advance at least one vaccine candidate initially targeting onchocerciasis in infants and children below 5 years of age, through Phase I human trials by 2025. Notably, *Ov*-RAL-2 and *Ov*-103 antigens have shown great promise during pre-clinical development, however, the overall success rate of vaccine candidates during clinical development remains relatively low due to certain adverse effects and immunogenic limitations. This study, thus, aimed at predicting the safety and immunogenicity of *Ov*-RAL-2 and *Ov*-103 potential onchocerciasis vaccine candidates prior to clinical trials. Advanced molecular simulation models and analytical immunoinformatics algorithms were applied to predict potential adverse side effects and efficacy of these antigens in humans. The analyses revealed that both *Ov*-RAL-2 and *Ov*-103 demonstrate favourable safety profiles as toxicogenic and allergenic epitopes were found to be absent within each antigen. Also, both antigens were predicted to harbour substantial numbers of a wide range of distinct epitopes (antibodies, cytokines, and T- Cell epitopes)

**Data Availability Statement:** All relevant data are within the manuscript and its Supporting information files.

**Funding:** The author(s) received no specific funding for this work.

**Competing interests:** The authors have declared that no competing interests exist.

associated with protective immunity against onchocerciasis. In agreement, virtual vaccination simulation forecasted heightened, but sustained levels of primary and secondary protective immune responses to both vaccine candidates over time. *Ov*-103 was predicted to be non-camouflageable, as it lacked epitopes identical to protein sequences in the human proteome. Indeed, both antigens were able to bind with high affinity and activate the innate immune TLR4 receptor, implying efficient immune recognition. These findings suggest that *Ov*-RAL-2 and *Ov*-103 can induce sufficient protective responses through diverse humoral and cellular mechanisms. Overall, our study provides additional layer of evidence for advancing the clinical development of both vaccine candidates against onchocerciasis.

## Introduction

Onchocerciasis or river blindness is a neglected tropical filarial ailment resulting from the parasitic activity of the nematode *Onchocerca volvulus* and transmitted through infected black flies of *Simulium* species. This disease poses a significant public health challenge, with a staggering over 205 million individuals worldwide residing in areas where parasite transmission is ongoing. The global incidence is estimated to reach approximately 20.9 million infections, impacting a substantial population of nearly 14.6 million individuals afflicted by skin-related complications, and an additional 1.15 million persons enduring vision impairment as a dire consequence of the infection in 2017 [1]. The magnitude of this burden has spurred the establishment of numerous expansive regional control and elimination initiatives, each employing distinct strategies to attain predefined targets. Notable advancements in disease elimination have been observed in Latin America, where four out of six endemic countries have declared onchocerciasis eliminated. Similar progress has been made in specific areas of Africa, such as in Mali and Senegal [2]. These successes have underscored the feasibility of broader elimination efforts throughout the African continent, mainly via the adoption of ivermectin-based Community-Directed Treatment (CDTi) [3, 4]–stimulating a shift in the global public health goal from control to elimination [5]. CDTi was a community-based approach where local populations took charge of distributing ivermectin to control onchocerciasis. This method empowered communities, leading to higher treatment compliance and sustainability. Nevertheless, increasing evidence suggests that relying solely on the CDTi and the existing control measures will be insufficient to achieve the elimination of onchocerciasis in Africa [6]. Additional strategies will be necessary to effectively combat the disease and achieve the desired elimination goals [7].

It is suggested that anthelmintic vaccines could be used as standalone tools or in conjunction with CDTi, through vaccine-linked chemotherapeutic programmes in achieving disease elimination [8]. Indeed, modelling studies predict that a suitable vaccine will be the most cost-effective means of onchocerciasis control [9, 10], as in addition to direct protection, a vaccine would mitigate anthelmintic drug resistance emergence while simultaneously reducing the chance of disease recrudescence [2]. Furthermore, substantial support exists for the notion that adopting such a strategy will safeguard the significant investments of current and previous onchocerciasis control programs. By implementing the suggested approach, the gains made through previous efforts can be preserved, reinforcing the overall effectiveness of the control and elimination endeavours [8]. Consequently, there is an urgent need for prophylactic or therapeutic vaccines to aid in the eradication of onchocerciasis. The implementation of such vaccines would provide a powerful and complementary tool to existing control measures,

enhancing the prospects of successfully eliminating the disease and preventing its resurgence in the future [11].

The viability of an *O. volvulus* vaccine and the rationale for screening potentially protective antigens find support in two key observations in onchocerciasis endemic regions. Firstly, a subset of individuals in the population, referred to as putative immunes (PI), neither exhibits patent infections (microfilaria positive) nor onchocerciasis-specific clinical manifestations of the disease [12–14]. Secondly, another group of individuals, as they age, develop concomitant immunity to parasitic microfilaria worms [15]. Indeed, much has been done to understand the mechanisms of acquired immunity to the infection in humans and it was established that the responses are largely linked to mixed Th1/Th2 responses against the *O. volvulus* larvae stage 3 (L3) of the parasite, and antibodies which together with cytokines induce antibody-dependent cell-mediated cytotoxicity (ADCC) [15, 16]. These correlates of protection served as a premise for the design and development of the current vaccine candidates, including *Ov*-RAL-2 and *Ov*-103.

The rigorous evaluation process for the selection and testing of *Ov*-RAL-2 and *Ov*-103 encompassed several stringent assessment criteria that have been reviewed elsewhere [9, 12, 17]. The decision to focus on these specific protein antigens was rooted in their proven ability to elicit a protective immune response, as demonstrated in both human studies and animal models of onchocerciasis [18–21]. The Onchocerciasis Vaccine for Africa (TOVA) Initiative was established in 2015, to advance at least one vaccine candidate initially targeting onchocerciasis in infants and children below 5-years of age, through Phase I human trials by 2025. Despite the potential promise of *Ov*-RAL-2 and *Ov*-103 as the leading TOVA candidates for clinical testing in humans, it is important to recognise that they are not exempt from certain limitations common to sub-unit protein-based biologics. Challenges such as high toxicity levels [22], allergenicity [23], poor innate immune-antigen recognition [24], and immune camouflage [25], particularly when targeting pathogens with immunomodulatory potentials like *O. volvulus*, often contribute to disappointing outcomes in human proof-of-concept clinical trials. To aid the onchocerciasis vaccine research community in addressing some of these challenges, we have employed immunoinformatics and molecular simulation models to predict the safety and immunogenicity of *Ov*-RAL-2 and *Ov*-103 potential onchocerciasis vaccine candidates prior to clinical trials [12, 17]. This approach allows us to model the efficacy of these antigens in humans before embarking on clinical studies. The results obtained provide an additional layer of evidence to guide the design and progression of these vaccine candidates through clinical development.

## Methodology

### Research design

This study was structured into three main immunoinformatics phases as a comprehensive approach to assess the safety and efficacy of *Ov*-RAL-2 and *Ov*-103 vaccine candidates from multiple immunological perspectives:

1. **Epitope Analysis:** diverse epitopes associated with key characteristics of the vaccine candidates were evaluated using different modelling tools, each with a unique set of parameters and individually trained on distinct experimental datasets, to identify immune-responsive epitopes representing each antigen's:

- Safety profile

- Immunomodulatory (immune camouflage) properties

- Potential for antibody-mediated (humoral) immune responses

- Capacity to elicit cellular immune responses

- Ability to induce cytokine-based responses, which mediate and bridge humoral and cellular immunity

2. **Innate Immune Recognition:** Using structural biology techniques, we investigated how effectively each vaccine candidate could be recognised by the innate immune system's TLR4 receptor

3. **Virtual Vaccination-Immunisation Simulation:** We conducted a computational simulation to model both primary and secondary immune responses over time. This included:

- Humoral (antibody-mediated) responses

- Cellular immune responses and,

- Cytokine-mediated responses

## Protein sequence and structure retrieval

The complete amino acid sequences of *Ov*-RAL-2 and *Ov*-103 (WormBase IDs OVOC9988 and OVOC4230, respectively) were obtained from WormBase (http://www.wormbase.org/) while that of TLR4 agonist was obtained from UniProtKB (ID P9WHE3) in the FASTA format. The TLR4 agonist used in this study is a 50S ribosomal *Mycobacterium tuberculosis* protein with the ability to stimulate TLR4 and was used in the docking and simulation sections as a molecular control adjuvant. The tertiary structure of human TLR4 (PDB ID: 4G8A) was retrieved from the Protein Data Bank, PDB (www.rcsb.org/pdb/home/home.do).

## Prediction of adverse reactions—Allergenicity and toxigenicity

The potential toxicity of the two vaccine candidates (*Ov*-RAL-2 and *Ov*-103) in humans was assessed using the ToxinPred server (http://crdd.osdd.net/raghava/toxinpred/), which generates overlapping epitopes from the input protein sequence. The ToxinPred server employs machine learning models and a quantitative matrix based on various peptide properties for the prediction of toxicogenic peptides in query protein sequences. The server is highly accurate, achieving an accuracy of 94.50% with a Matthews Correlation Coefficient (MCC) of 0.88 [22]. Additionally, for allergenicity prediction, the AllerTOP v2.0 and AllergenFP servers were employed. AllerTOP v2.0 (http://www.ddg-pharmfac.net/AllerTOP) utilizes amino acid E-descriptors, auto- and cross-covariance transformation, and k nearest neighbours (kNN) machine learning methods for allergen classification, displaying an accuracy of 85.3% in a 5-fold cross-validation process [26]. Conversely, AllergenFP (http://ddg-pharmfac.net/AllergenFP/) utilizes an alignment-free, descriptor-based fingerprint approach to distinguish allergens from non-allergens. Applied to allergens versus non-allergenic peptides, this method accurately identified 88% of them with a Matthews correlation coefficient of 0.759 [23].

## Cross-conservation analysis for assessment of immune camouflage

The BLAST algorithm (https://www.uniprot.org/blast/) from UniProtKB was employed to predict potential cross-reactivity between each vaccine candidate and the human proteome, focusing on shared T-cell-receptor-interacting residues of putative T-cell epitopes post-matching procedure. The sequences of both vaccine candidates, *Ov*-RAL-2 (OVOC9988) and *Ov*-

103 (OVOC4230) were used as inputs. BLASTp parsed and searched the UniProt-reviewed human proteome dataset to identify identical sequence stretches common to both the input and human sequences. Studies have shown that pathogenic peptides with matches to sequence stretches in the human proteome tend to bind to the same HLA allele, suggesting a potential bias towards immune tolerance or camouflage [27]. For the BLAST search of the human proteome database, customized parameters were implemented, including a 0.01 E-threshold, BLOSUM62 matrix, filtered low complexity regions, and exclusion of gapped regions. The E-threshold value is a statistical measure representing the number of expected matches in a random dataset, where a smaller E-threshold indicates a more significant match [28]. The BLOSUM matrix provides probability scores for alignments based on substitution frequencies within related proteins [29].

## Assessment of humoral immune epitopes

**Antigenicity.** The antigenicity of the two vaccine candidates was predicted using two alignment-free servers, ANTIGENpro and VaxiJen v2.0. ANTIGENpro (http://scratch. proteomics.ics.uci.edu/) utilizes protein antigenicity microarray data to forecast protein antigenicity and has shown an estimated accuracy of 76% in cross-validation experiments [30]. On the other hand, the VaxiJen 2.0 server (http://www.ddg-pharmfac.net/vaxijen/VaxiJen/ VaxiJen.html) employs auto- and cross-covariance (ACC) transformation of protein sequences, converting them into uniform vectors of principal amino acid properties for antigenicity prediction [31].

**Ig-class prediction.** The involvement of specific Ig-classes and subclasses in immune protective responses against *O. volvulus* parasites has been reported in previous studies [32, 33]. Differences in levels of *Ov*-specific IgG, IgG subclasses, and IgE have been observed between infected and putatively immune individuals [34]. To predict B-cell epitopes capable of inducing specific classes of antibodies (IgG, IgE, and IgA), the IgPred server (http://crdd.osdd.net/ raghava/igpred/) was utilized with the default threshold of 0.9. The server utilizes models developed for predicting antibody class-specific B-cell epitopes, incorporating features such as amino acid composition, dipeptide composition, and binary profiles [35].

**Linear B-cell epitopes.** These were predicted using the BepiPred-2.0 web server (http:// www.cbs.dtu.dk/services/BepiPred/), which relies on a random forest algorithm trained on epitopes annotated from antibody-antigen protein structures. This method demonstrated superiority over other available tools in sequence-based epitope prediction, encompassing epitope data from solved 3D structures and a vast collection of linear epitopes sourced from the IEDB database [36]. In addition, both vaccine candidates were subjected to linear epitope prediction utilizing two other servers. Initially, ABCpred was used to predict 16-mer epitopes at a set threshold of 0.70 based on recurrent neural networks (http://www.imtech.res.in/raghava/ abcpred/) [37]. Subsequently, the BCPreds server (http://ailab-projects2.ist.psu.edu/bcpred/ predict.html), an artificial intelligence-based novel tool was employed to predict 20-mer non-overlapping epitopes set at a default specificity threshold of 0.75.

**Discontinuous B-lymphocyte (DBL) epitope prediction.** More than 90% of B-cell epitopes in nature are estimated to be discontinuous [38]. To predict these discontinuous (conformational) B-cell epitopes for the validated 3D structures produced as described in the structure prediction section below, the ElliPro tool (http://tools.iedb.org/ellipro/) was utilized. ElliPro incorporates three algorithms: approximating the protein shape as an ellipsoid, calculating the residue protrusion index (PI), and clustering neighbouring residues based on their PI values. The output epitopes are scored with a PI value averaged over each epitope residue, where an ellipsoid with a PI value of 0.9 includes 90% of the protein residues, leaving 10%

outside the ellipsoid. The PI value for each epitope residue is determined based on the center of mass of residues residing outside the largest possible ellipsoid. Among various structure-based methods for epitope prediction, ElliPro outperformed others, yielding an AUC value of 0.732 for the most significant prediction for each protein [39].

## Assessment of cellular immune epitopes

**Cytotoxic T lymphocyte (CTL) epitopes.** To predict CTL epitopes for both vaccine candidates, the NetCTL 1.2 server (http://www.cbs.dtu.dk/services/NetCTL/) was employed. This server integrates the prediction of MHC class I binding peptides, proteasomal C-terminal cleavage sites, and TAP (Transporter Associated with Antigen Processing) transport efficiency. The assessments of MHC class I binding and proteasomal cleavage are carried out using artificial neural networks. For this study, our focus was on the A2, A3, A24, and B7 MHC class I supertypes, as these offer optimal population coverage for onchocerciasis endemic zones [40]. The CTL epitope prediction was conducted with the default threshold value of 0.75 for all the selected supertypes.

**Helper T-cell (HTL) epitope prediction.** CD4+ T-lymphocyte epitopes play a crucial role in stimulating robust protective immune responses during peptide-based vaccination [41]. Additionally, they have been recognized as significant contributors to immune protective mechanisms against *Onchocerca* species [42, 43]. To identify these helper T-cell (HTL) epitopes for human alleles, the NetMHCII 2.3 Server (http://www.cbs.dtu.dk/services/NetMHCII/) was utilized. The NetMHCII 2.3 server leverages artificial neural networks to predict 15-mer peptides binding to specific HLA-DR, HLA-DQ, and HLA-DP alleles [44].

## Assessment of cytokine-inducing epitopes (IFN-γ, IL-4, IL5, IL-17, IL-10 and TNF-α)

Interferon-gamma (IFN-γ) is vital in both adaptive and innate immune responses [45] and has been reported as a significant marker for Th1 responses in onchocerciasis protection [34]. IFN-γ 15-mer epitopes for both vaccine candidates were predicted using the IFNepitope server (http://crdd.osdd.net/raghava/ifnepitope/scan.php). This server uses overlapping sequences constructed from the input sequence to predict IFN-γ epitopes. It relies on a dataset of IFN-γ inducing and non-inducing MHC class-II binders capable of activating T-helper cells [46], with prediction carried out using a hybrid motif and support vector machine (SVM) models. IL-4 is characteristic of T-helper 2 responses, produced by CD4+ T cells in response to helminths [47]. Both IL-4 and IL-5 are implicated in protective responses against *Onchocerca* species [48]. 15-mer IL-4 inducing epitopes from both vaccine candidates were predicted using the IL4pred webserver (http://crdd.osdd.net/raghava/il4pred/) with an SVM default threshold of 0.2. Overlapping sequences are used for prediction, based on a dataset containing 904 experimentally validated IL4-inducing and 742 non-inducing MHC class II binders, achieving an accuracy of 75.76% and a Matthews's Correlation Coefficient (MCC) value of 0.51. Since IL-5 has been reported to be implicated in the generation of protective immune responses, 15-mer IL-5-inducing epitopes were predicted for both vaccine candidates using the default hybrid model probability threshold of 0.2 on the IL5Pred server (https://webs.iiitd.edu.in/raghava/il5pred/index.html). The server is based on models (including eXtreme Gradient Boosting and RF-based) that have been trained, tested, and validated on experimentally validated 1907 IL-5-inducing and 7759 non-IL-5-inducing peptides obtained from IEDB with the hybrid method achieved AUC 0.94 with MCC 0.60 on a validation/independent dataset [49].

In addition to IL-4 and IL-5, Th17 helper cells are reported to be involved in hyperreactive onchocerciasis in infected individuals [43]. Interleukin-17 (IL-17) is a key player in the

mammalian immune system, exerting a host-defensive role in infectious diseases [50]. IL-17 15-mer inducing peptides for both vaccine candidates were predicted using the IL17eScan server (http://metagenomics.iiserb.ac.in/IL17eScan/) with the DPC model for enhanced accuracy. The default threshold of 0.6 was used for epitope prediction. The server's dataset comprises 338 IL-17-inducing and 984 IL-17 non-inducing peptides from the Immune Epitope Database. The dipeptide composition-based SVM model demonstrated 82.4% accuracy with Matthews correlation coefficient = 0.62 at polynomial (t = 1) kernel on 10-fold cross-validation, outperforming RF [51].

Immunosuppression remains a major concern in vaccine development against helminths, including onchocerciasis, with female adult worms capable of living within human hosts for extended periods [52]. Filariasis animal model studies indicate that the downregulation of the immune response is dependent on IL-10 mediated by female worms [53]. 15-mer IL-10-inducing epitopes were predicted using the Random Forest model of the IL-10pred tool (http://crdd.osdd.net/raghava/IL-10pred/). The server constructs overlapping sequences for IL-10-inducing MHC II binders' prediction. Based on a dataset of 394 experimentally validated IL-10-inducing and 848 non-inducing peptides, the Random Forest-based model achieved a maximum Matthews's Correlation Coefficient (MCC) value of 0.59 with an accuracy of 81.24%, using dipeptide composition [54].

Lastly, studies in mice models have suggested TNF-$\alpha$ as one of the soluble immune factors that might be associated with vaccine-induced protection, based on the analyses of diffusion chamber fluid of mice immunized with *Ov*-FUS-1 [20]. The TNFepitope server (https://webs.iiitd.edu.in/raghava/tnfepitope/) was used to predict human TNF-$\alpha$-inducing epitopes in *Ov*-103 and *Ov*-RAL-2 based on the default threshold of 0.45. The prediction models for the server were trained and validated using the experimentally validated TNF-$\alpha$ inducing/non-inducing epitopes from human and mouse hosts with the hybrid model (combination of alignment-free and alignment-based method) achieving a maximum AUROC of 0.83 and 0.77 on independent datasets for human and mouse hosts, respectively [55].

## Virtual immunisation simulation

To explore the immunogenicity and immune response profiles of both vaccine candidates, *in silico* immune simulations were performed using the C-ImmSim server (http://150.146.2.1/C-IMMSIM/index.php). C-ImmSim is an agent-based model that employs a position-specific scoring matrix (PSSM) for predicting immune epitopes and utilizes machine learning techniques to simulate immune interactions. The server replicates three distinct compartments found in mammals: (i) the bone marrow, where hematopoietic stem cells produce new lymphoid and myeloid cells; (ii) the thymus, where naïve T cells are selected to prevent autoimmunity; and (iii) a tertiary lymphatic organ, such as a lymph node [56]. Following the target product profile of a prophylactic onchocerciasis vaccine proposed by The Onchocerciasis Vaccine for Africa (TOVA) Initiative, we administered three injections at four-week intervals [11]. The default parameters were employed for the simulation, with time steps set at 1, 84, and 168 (each representing 8 hours, and time step 1 corresponds to the injection at time = 0). Thus, three injections four weeks apart were administered. Additionally, to simulate repeated exposure to each antigen, mimicking conditions in an endemic area, we administered 12 injections of both proteins separately at four-week intervals to explore for clonal selection. The Simpson index, D (a measure of diversity), was analysed from the plot to assess the results.

## Structural prediction and assessment of innate immune recognition of vaccine candidates

**Structure prediction.** Secondary structure and disorder prediction was performed using the PSIPRED server (http://bioinf.cs.ucl.ac.uk/psipred/) and the RaptorX Property web server (http://raptorx.uchicago.edu/StructurePropertyPred/predict/) [57]. PSIPRED utilizes two feed-forward neural networks and PSI-BLAST output to obtain reliable secondary structure predictions, while RaptorX Property employs DeepCNF, a machine learning model, to concurrently predict secondary structure, solvent accessibility, and disordered regions [57, 58]. The RaptorX Property server achieved high accuracy for secondary structure and disorder prediction. Furthermore, the IUPred3 server (https://iupred3.elte.hu/) was used to predict intrinsically disordered regions (IDRs) in both vaccine candidates, as IDRs can serve as structural antigens and are found in leading vaccine candidates [59, 60]. For tertiary structure prediction, homology modelling was performed using the RaptorX server (http://raptorx.uchicago.edu/), which delivers high-quality structural models by considering the alignment between a target sequence and template proteins, nonlinear scoring, and probabilistic-consistency algorithms [61]. The predicted 3D models were then refined using the ModRefiner server (https://zhanglab.ccmb.med.umich.edu/ModRefiner/) and the GalaxyRefine server (http://galaxy.seoklab.org/cgi-bin/submit.cgi?type=REFINE). ModRefiner performs atomic-level energy minimization to improve global and local structures, while GalaxyRefine rebuilds side chains and performs molecular dynamics simulation for overall structure relaxation, resulting in enhanced local structure quality [62, 63]. To validate the tertiary structures, the ProSA-web server (https://prosa.services.came.sbg.ac.at/prosa.php) was used to calculate an overall quality score and compare it to known protein structures. The ERRAT server (http://services.mbi.ucla.edu/ERRAT/) analysed non-bonded atom-atom interactions, and the RAMPAGE server (http://mordred.bioc.cam.ac.uk/~rapper/rampage.php) provided a Ramachandran plot to assess the model's quality based on the percentage of residues in allowed and disallowed regions [64, 65].

**Data-driven docking of vaccine candidates with TLR4 receptor.** To predict how the vaccine candidates interact with the innate immune system via TLR4 binding, we conducted data-driven docking on the (High Ambiguity Driven protein-protein DOCKing (HADDOCK) server (https://milou.science.uu.nl/services/HADDOCK2.2/haddockserver-easy.html). HADDOCK employs Python scripts utilising Crystallographic and NMR structures produced experimentally for structure calculations [66, 67]. HADDOCK combines experimental and/or theoretical data to optimize docking and enhance scoring and sampling accuracy. We identified active and passive residues for each vaccine candidate (*Ov*-103, *Ov*-RAL-2, or TLR4 agonist) using the CPORT server (Consensus Prediction Of interface Residues in Transient complexes) (https://milou.science.uu.nl/services/CPORT/). The CPORT server identified active and passive amino acid residues essential for interaction with the TLR4 receptor, including Phe31, Ile34, Lys35, Phe38, Glu106, Leu156, Arg157 for *Ov*-103; Asp34, Thr64, Asp65, Gln66, Glu69, Val94, Ala97, Arg98, Tyr101 for *Ov*-RAL-2; and Thr35, Ala36, Ala38, Pro39, Val40, Ala41, Val42, Ala43, Ala44, Ala45, Gly46, Ala47, Ala48, Pro49 for the TLR4 agonist. The TLR4 agonist (control antigen) used in this study is a Mycobacterium tuberculosis 50S ribosomal protein (UniProtKB ID P9WHE3) known to have a high affinity for the TLR4 receptor. The active residues of the TLR4 receptor (PDB ID: 4G8A) were identified from a solved lipopolysaccharide X-ray co-crystalized model of the receptor [68]. These TLR4 active residues include Arg264, Lys341, Lys362, Lys388, Asn417, Glu439, Phe440, and Phe463 [68]. Subsequently, the 3D structures of each vaccine candidate and TLR4 were submitted to the HADDOCK server as ligand and receptor, respectively, for the docking calculations. Based on

the lowest HADDOCK scores and biological rationale, we identified top-ranked clusters and representative structures of TLR4-vaccine complexes and visualised them using Pymol.

**Molecular dynamics (MD) simulation of vaccine candidates in complex with TLR4 receptor.** Molecular dynamics (MD) simulations were conducted to evaluate the stability of the vaccine-TLR4 complexes using GROMACS version 2018 [69] with the Optimized Potential for Liquid Simulation (OPLS-AA) force field [70]. Initially, all systems were solvated with the TIP4P water model [71] in cubic boxes and neutralized by adding sodium or chloride ions. Two energy minimization steps employing the steepest descent and conjugate gradient algorithms were performed for 50,000 steps to eliminate unfavorable contacts, and the systems converged at Fmax < 1000 kJ/mol. The minimized systems underwent a two-step equilibration process with position restraints on all atoms. Firstly, the systems were equilibrated in the NVT ensemble (constant Number of particles, Volume, and Temperature) at 300 K for 100 ps using the V-rescale method. Subsequently, they were subjected to an NPT ensemble (constant Pressure) with the Perrinello-Rahmna barostat at an isotropic pressure of 1 bar for 500 ps. After equilibration, MD production runs were performed without position restraints for 40 ns to 60 ns, depending on the system, in the NPT ensemble. During the production step, both temperature and pressure were maintained at 300 K and 1 bar, respectively, using the V-rescale thermostat [72] and Perronello-Rahman barostat [73]. For all systems, periodic boundary conditions (PBC) were applied in all directions, and the Particle Mesh Ewald (PME) method [74] was used for long-range electrostatic interactions with a cutoff distance of 0.1 nm for both van der Waals and electrostatic interactions. The LINCS algorithm [75] was employed to constrain bond lengths, and a time step of 2 fs was used for all simulations.

**Binding affinity and dissociation constant estimation.** To evaluate the binding affinity of the vaccine candidates to TLR4 receptor, 20 snapshots of each vaccine complex were extracted at a predetermined time of 2 ns, and the time-lapse binding affinity was assessed using the PRODIGY (PROtein binDIng enerGY prediction) webserver which predicts the binding affinity of protein-protein complexes from their 3D structures [76]. Additionally, the affinities were estimated based on the equation: $\Delta G = -RT \, lnk_d$; where $\Delta G$ represents the estimated binding affinity, R the universal gas constant, T temperature (25 ºC), and $k_d$ the dissociation constant.

# Results

## Research design

This study was designed to, firstly, assess various epitopes which in each vaccine candidate represent responsiveness to vaccine candidates' potential adverse side reactions, immunomodulation (camouflage) property, antibody-based (humoral) responses, cellular immune responses, as well as cytokine-based responses (regulation and mediation between humoral and cellular responses). Different modelling tools with a unique set of parameters and individually trained on distinct experimental datasets were used to achieve this. Secondly, through structural simulations, the study assessed the ability of each vaccine candidate to be recognised by the innate immune TLR4 receptor. Finally, a virtual vaccination simulation was also performed to model the primary and secondary humoral, cellular, and cytokine-mediated responses over time.

## Toxigenicity and allergenicity predictions indicate that both *Ov*-RAL-2 and *Ov*-103 are safe—Harbouring no epitopes responsive to adverse side effects

To ensure the safety of the peptide-based vaccine candidates, toxicity and allergenicity levels were predicted for both antigens. Toxic proteins or peptides have been found to induce

dementia (e.g Alzheimer's disease) and heart disease in some patient groups [77, 78]. The ToxinPred server was employed to identify potentially toxigenic epitopes, however, both antigens were predicted to have no toxic peptides or signals. ToxinPred simply determines whether or not a peptide is toxic based on machine learning models that were benchmarked on a dataset including 1805 toxic peptides [22], yielding a model accuracy of 94.50% with MCC 0.88. Additionally, two allergen estimator matrices, AllerTOP v.2 and AllergenFP servers indicated that both *Ov*-RAL-2 and *Ov*-103 were non-allergenic. The AllerTOP [26] method uses a k-nearest neighbour algorithm trained on a dataset of 2,427 known allergens and 2,427 non-allergens for classifying potential allergens, while the AllergenFP [23] approach utilises Tanimoto descriptor fingerprints of epitopes in vaccine candidates, classifying them against fingerprints of known allergenic peptides. *In vitro* experimental assays [12, 79] measuring IgE-related reactions also supports these findings, as samples from infants 1–5 years of age showed no adverse IgE responses to both vaccine candidates.

## *Ov*-RAL-2 and *Ov*-103 show negligible cross-conservation with the human proteome

To assess potential tolerogenic responses, T-cell epitopes in *Ov*-RAL-2 and *Ov*-103 were evaluated to identify cross-conserved epitopes in common with the human proteome. UniProt BLAST was used for this analysis. Whereas a few T-cell epitopes in *Ov*-RAL-2 matched peptides in some human proteins (UniProtKB IDs: Q92896, P20265, and P42858), *Ov*-103 showed no homology with the human proteome, making it potentially non-camouflageable. Further analysis of the *Ov*-RAL-2 stretches matching human sequences revealed (Fig 1) that a low complexity Glutamine Repeat Region (GRR) motif accounted for most of the observed sequence identity with similar GRR regions in different human proteins. However, the GRR region only represented 6.7% of the full length of *Ov*-RAL-2, suggesting that any tolerogenicity could very much likely be negligible.

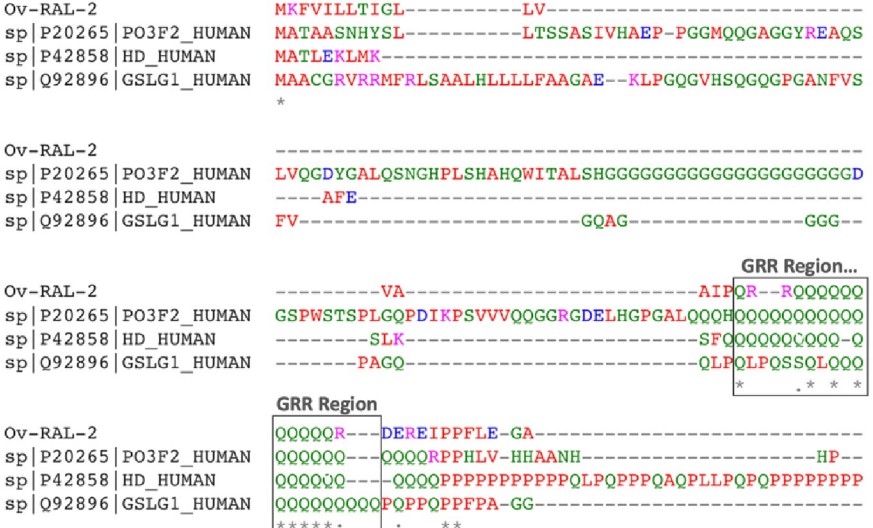

**Fig 1. Multiple sequence alignment of *Ov*-RAL-2 Glutamine-repeat epitope (GRR) showing identical sequence stretches common to some human proteins.** UniProtKB IDs of top human hits Q92896 and P20265 function as transcription factors that play a key role in neuronal differentiation, while P42858 plays a role in microtubule-mediated transport or vesicle function with its N-terminal fragment involved in the formation of autophagic vesicles.

## Assessment of humoral epitope content of vaccine candidates

**Prediction of antigenicity of Ov-RAL-2 and Ov-103 in humans.**   To assess the antigenicity of both antigens, we employed two servers: VaxiJen 2.0 and ANTIGENpro, with the target organism set to "parasite." *Ov*-RAL-2 showed a score of 0.5195 on VaxiJen and 0.887977 on ANTIGENpro. While, *Ov*-103 received scores of 0.5234 on VaxiJen and 0.594738 on ANTIGENpro. Antigenicity scores >0.5 threshold value for both VaxiJen [31] and ANTIGENpro [30] servers are considered significant. Overall, although the antigenicity scores for both antigens were at the borderline of threshold of significance (0.5), it is crucial to acknowledge that high antigenicity levels for both antigens have been demonstrated in early-stage preclinical and human research [18, 19]. These studies revealed a strong correlation between antibody reactions to *Ov*-103 and *Ov*-RAL-2 and protective immunity against *Onchocerca volvulus* in mice and humans. This is consistent with other research findings where antigens predicted to have low to moderate *in silico* antigenicity led to significantly ($p < 0.0001$) high IgG antibody responses against the modelled antigen when tested in mice [74]. Therefore, according to VaxiJen and ANTIGENpro scores, *Ov*-RAL-2 and *Ov*-103 proved to be significantly antigenic.

**Linear and discontinuous B-cell epitope prediction.**   Linear B-cell epitopes were predicted for *Ov*-RAL-2 and *Ov*-103 using three servers: ABCpred, BCPreds, and BepiPred 2.0. For *Ov*-103, the ABCpred, BCPreds, and BepiPred servers identified 10, 9, and 3 linear B-cell epitopes respectively (S1 File). Meanwhile *Ov*-RAL-2 had 19, 44, and 3 epitopes, respectively (S2 File). Unlike linear B-cell epitopes which refer to specific amino acid sequences within an antigen, discontinuous (conformational) B-cell epitopes are formed by the three-dimensional structure of the antigen, creating a unique complementary surface for B-cell binding to trigger antibody responses. For *Ov*-103, a total of 67 residues were predicted to be found in five different discontinuous epitopes, with scores ranging from 0.683 to 0.944 (Table 1). These conformational epitopes varied in size, spanning from 3–26 residues. Likewise, for *Ov*-RAL-2, six

**Table 1. Amino acid residues predicted to be found in conformational B-lymphocyte epitopes.**

| Antigen | Sets of conformational epitope residues | Size (aa) | Score* |
|---|---|---|---|
| *Ov*-RAL-2 | :A15, _:A16, _:I17, _:P18, _:Q19, _:R20, _:Q22, _:Q23, _:Q24, _:Q25, _:Q26,: Q27, _:Q28, _:Q29, _:Q30, _:Q31, _:R33, _:D34 | 18 | 0.805 |
| | :A99, _:D100, _:R103, _:I104, _:Q107, _:A108, _:V109, _:A110, _:R111, _:F112, _:S113, _:P114, _:A115, _:A116, _:K117, _:D118, _:R122, _:A143, _:D146, _:S147, _:L148, _:S149, _:E150, _:S151, _:V152, _:R153, _:R154, _:E155 | 28 | 0.732 |
| | :A70, _:E73, _:A74, _:N77, _:R78, _:L79, _:G80, _:G81, _:S82, _:Y83, _:K84,: V85, _:R86, _:T88, _:Q89, _:E92, _:E93, _:K96 | 18 | 0.716 |
| | :S161, _:P162, _:Q163, _:E164 | 4 | 0.691 |
| | _:N62, _:K63, _:T64, _:Q66, _:Q67, _:S129, _:P130, _:H131, _:L132, _:T133,: Q136 | 11 | 0.562 |
| | :E43, _:G44, _:A45, _:P46 | | 0.532 |
| *Ov*-103 | :T146, _:C147, _:I148, _:V149, _:P150, _:V151, _:L152, _:I153, _:N154,: T155, _:L156, _:R157 | 12 | 0.944 |
| | :D41, _:E42, _:K43 | 3 | 0.867 |
| | :F140, _:A141, _:T142, _:Y143, _:L144, _:F145 | 5 | 0.723 |
| | _:F31, _:T32, _:D33, _:K35, _:F38, _:A39, _:K40, _:Q44, _:Q47, _:S90, _:R92, _:E93, _:T94, _:M95, _:S96, _:N97, _:P98, _:K99, _:M100, _:D101,: F102, _:T103, _:N104, _:K105, _:E106, _:N107 | 26 | 0.704 |
| | _:M1, _:Q67, _:P68, _:A70, _:N71, _:D72, _:M73, _:Q74, _:K75, _:T76, _:G79, _:K80, _:G82, _:D83, _:S86, _:V118, _:T119, _:E120, _:G121, _:K124 | 20 | 0.683 |

*Threshold score for epitope prediction is 0.50

**Table 2. Abundance of different epitope types in *Ov*-103 and *Ov*-RAL-2.**

| Protein | CTL | HTL | IL-4 | IL-5 | IFN-γ | IL-17 | IL-10 | TNF-α | IgG | IgA | LBL | DBL |
|---------|-----|-----|------|------|-------|-------|-------|-------|-----|-----|-----|-----|
| *Ov*-103 | 20 | 53 | 111 | 61 | 20 | 33 | 82 | 19 | 15 | 12 | 22 | 5 |
| *Ov*-RAL-2 | 15 | 123 | 54 | 29 | 58 | 3 | 87 | 32 | 4 | 0 | 66 | 6 |

CTL: Cytotoxic T-cell epitope, HTL: Helper T-cell epitope, IL4–17: Interleukin 4–17, IFN-γ: Interferon-γ, IgG–A: immunoglobulinG–A,

LBL: Linear B-Cell, and DBL: Discontinuous B-Cell epitopes, TNF-α: Tumor necrosis factor-α

different discontinuous epitopes comprising 73 residues were predicted (Table 2), with scores ranging from 0.532 to 0.805, with a threshold value of 0.50 for both RAL-2 and *Ov*-103. These conformational epitopes (*Ov*-RAL-2) consisted of 4 to 28 residues.

**Immunoglobulin (Ig) inducing epitopes.** To predict the presence of specific epitopes capable of inducing immunoglobulin (Ig) production, the IgPred web server was employed to analyse the protein sequences. The analysis revealed (Table 2, S1 and S2 Files) a total of 27 Ig-inducing epitopes in *Ov*-103, with 12 predicted to induce IgA production and the remaining ones promoting IgG production. On the other hand, *Ov*-RAL-2 was found to have four predicted epitopes capable of inducing IgG. Although the tool was unable to differentiate between IgG isotypes, the presence of predicted IgG epitopes in both proteins suggests an intrinsic property to potentially trigger an appropriate immune response involving Th1 and/or Th2 immunity in humans. These findings align with extensive research conducted on animal models and human samples, emphasising the importance of a mixed Th1/Th2 response [12] in targeting onchocerciasis through vaccination.

## Assessment of cellular epitope content of vaccine candidates

**Helper T (CD4+) and Cytotoxic T lymphocyte (CD8+) epitope prediction.** To identify potential CTL epitopes, we utilized the NetCTL 1.2 server and considered the HLA-A2, A3, A24, and B7 supertypes, which together are known to have broad peptide-binding specificities [80]. Due to the broad coverage attained, these supertypes are of particular interest in designing vaccines that can be effective across diverse human populations, as they have the potential to present epitopes from various pathogens to CD8+ lymphocytes [80, 81]. Our predictions indicate that (Table 2, S1 and S2 Files) *Ov*-103 and *Ov*-RAL-2, respectively, contain 20 and 15 CTL epitopes (averagely 9-mers), binding across the selected supertypes. Notably, most predicted epitopes for both antigens were specific to the HLA-A2 supertype. CTL epitope predictions for both candidate antigens exhibited scores above the default threshold value of 0.75, resulting in a sensitivity of 80% and specificity of 97 [82]. On the other hand, we employed the NetMHCII 2.3 server to predict 15-mer overlapping HTL epitopes for *Ov*-103 and *Ov*-RAL-2, with specific binding to HLA-DR, HLA-DQ, and HLA-DP alleles. The predicted epitopes were classified as strong binders (SB) or weak binders (WB) based on their binding affinity. In both antigens, the SB epitopes' binding affinity values ranged from as low as $IC_{50}$ 9nM–1.3mM. For *Ov*-RAL-2 (Table 2, S2 File), a total of 123 SB was predicted (HLA-DR: 36, HLA-DQ: 51, HLA-DP: 36), while *Ov*-103 (Table 2, S1 File) yielded 53 strong binders (HLA-DR: 16, HLA-DQ: 14, HLA-DP: 23). The large number of CD4+ epitopes plus low $IC_{50}$ values for the predicted peptides in both proteins indicate that *Ov*-103 and *Ov*-RAL-2 possess good antigenic/immunogenic properties, capable of eliciting appropriate helper T cell responses.

## Assessment of cytokine-inducing epitope content of vaccine candidates

**Cytokine-inducing epitope prediction for IL-4, IL-5, IL-10, IL-17, IFN-γ, and TNF-α.** To identify IL-4-inducing epitopes, the IL4pred server was employed, resulting (Table 2, S1 and S2 Files) in the prediction of 54 IL-4 inducers for *Ov*-RAL-2 and 111 for *Ov*-103. These abundant IL-4-inducing epitopes suggest that both proteins can stimulate the desired Th2 responses required for protection against *O. volvulus* parasites in the human host. For IL-5, a total of 29 inducing epitopes were predicted to be present in *Ov*-RAL-2 while a total of 61 potential epitopes were predicted to be present in *Ov*-103 (Table 2, S1 and S2 Files). For *Ov*-RAL-2 and *Ov*-103, a total of 156 and 150 potential IFN-γ -inducing epitopes (15-mers) were predicted, respectively. However, only 58 epitopes for *Ov*-RAL-2 and 20 for *Ov*-103 scored (Table 2, S1 and S2 Files) significant values as predicted by the server. Studies on other filarial parasites, such as lymphatic filariasis, have suggested that IL-17 may have both protective and pathogenic effects [83]. IL-17eScan server was used to predict IL-17-inducing epitopes for *Ov*-RAL-2 and *Ov*-103. Three epitopes were predicted for *Ov*-RAL-2, while 33 epitopes were identified in *Ov*-103 (Table 2, S1 and S2 Files). In the context of vaccine protection, the presence of IL-17 epitopes suggests that both antigens have the potential to trigger protective immune responses against *O. volvulus* microfilariae within the human host. IL-10 is known to maintain an immunosuppressed environment in onchocerciasis [84]. The IL-10pred server was used to predict IL-10-inducing epitopes, resulting in the identification of 82 epitopes for *Ov*-103 and 87 for *Ov*-RAL-2 (Table 2, S1 and S2 Files). Lastly for TNF-α, a total of 32 potential epitopes were predicted to be present in *Ov*-RAL-2 while a total of 19 potential epitopes were predicted to be present in *Ov*-103 (Table 2, S1 and S2 Files).

## Virtual immunisation simulation: Humoral, cytokine, and cellular-based responses to *Ov*-103 and *Ov*-RAL-2 over time

We also conducted virtual immune simulations using the C-ImmSim machine learning-based tool to assess the humoral and cellular immune responses induced by both *Ov*-103 and *Ov*-RAL-2 antigens over time. Virtual immunisation simulation serves as a valuable tool in the early stages of vaccine development, aiding to model the nature of the human immune response to an antigen as a function of time, providing a wealth of information to guide the design and conduct of clinical trials. Based on the target product profile of a prophylactic onchocerciasis vaccine proposed by The Onchocerciasis Vaccine for Africa (TOVA) initiative, we simulated the administration of three injections at four-week intervals [11, 85]. The results revealed consistent and robust immune responses with distinct patterns in the primary, secondary, and tertiary response categories (Fig 2).

During the primary response (Fig 2a and 2b), both antigens elicited high levels of IgM antibodies, indicating the initial activation of the immune system. As the immune system virtually encountered the antigens repeatedly, the secondary and tertiary responses showed a significant increase in the levels of IgG1+IgG2, IgM, and IgG+IgM antibody levels, coupled with a gradual decrease in antigen levels (Fig 2a and 2b). This trend indicates the development of immune memory and the establishment of a more effective immune response upon subsequent exposures to the antigens. Furthermore, the data (Fig 2a and 2b) demonstrated that the levels of IgG1 for both candidate antigens were higher than those of IgG2 antibodies over time. For *Ov*-RAL-2, the level of IgM decreased sharply, after the third injection, than that of IgG1+IgG2 which was more sustained over time (Fig 2b). This shift in antibody production is indicative of a transition from the early humoral response dominated by IgM to a more specific and long-lasting response characterized by the IgGs. Meanwhile, *Ov*-103 showed higher and sustained levels of IgM than IgG1+IgG2 (Fig 2). However, IgM+IgG portrayed good and lasting levels

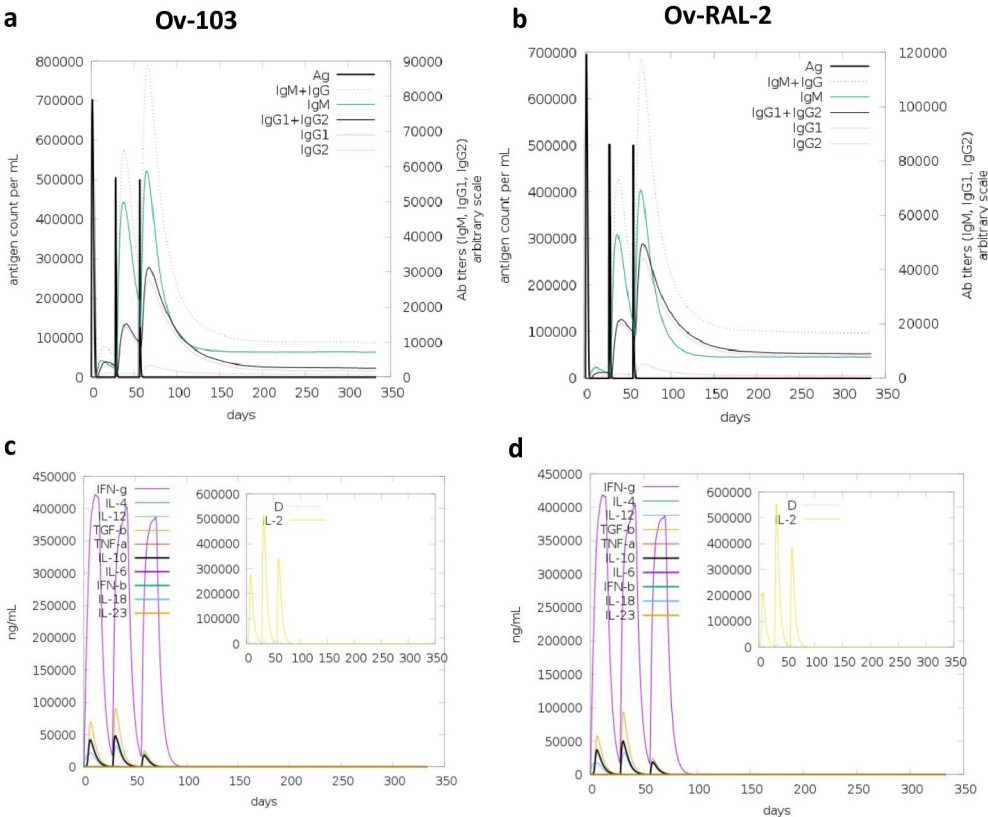

**Fig 2. Virtual immune response of antibodies and cytokine responses for both vaccine candidates.** Panels a) *Ov*-103 and b) *Ov*-RAL-2 indicate virtual immune responses for both antigens. Immunoglobulin production in response to antigen injections (black vertical lines); specific subclasses are indicated as coloured peaks. Panels c) *Ov*-103 and d) *Ov*-RAL-2 show virtual cytokine profiles for both candidate antigens with three injections given 4 weeks apart. The larger plot shows cytokine levels after the injections. The insert plot shows the IL-2 level with the Simpson index, D, indicated by the dotted line. D is a measure of diversity. An increase in D over time indicates the emergence of different epitope-specific dominant clones of T-cells. The smaller the D value, the lower the diversity.

for this antigen. The immune memory was also evident in the high levels of IFN-γ concentration and T helper cell population (Th2 response) throughout the exposure period (Fig 2c and 2d). IFN-γ is an essential cytokine produced by T cells, and its high levels indicate a persistently activated Th1 immune response.

Additionally, the simulations indicate (Fig 3) a high response in B-cell populations, as well as both TH (helper) and TC (cytotoxic) cell populations, further supporting the development of immune memory for both antigens. Generally, both antigens had similar levels of immune cell responses. The total B cell populations (Fig 3a and 3d) for both cases levelled at about 340 B cells per mm$^3$, and the total TH cell population for the two antigens drastically reduced at around day 75 to an estimated 200 memory TH cells per mm$^3$ (Fig 3b and 3e); while the concentration of T cytotoxic cell population was almost constant across 350 days from the point of antigen inoculation (Fig 3c and 3f). To simulate repeated exposure to the antigens that mimicked a typical endemic area context, we administered 12 injections. Overall, the simulations predicted slightly higher levels of antibodies against *Ov*-103 compared to *Ov*-RAL-2. These findings were consistent with our earlier predictions (Table 2) of Ig-inducing epitopes, where *Ov*-103 was shown to contain more epitopes than *Ov*-RAL-2. Both antigens demonstrated the

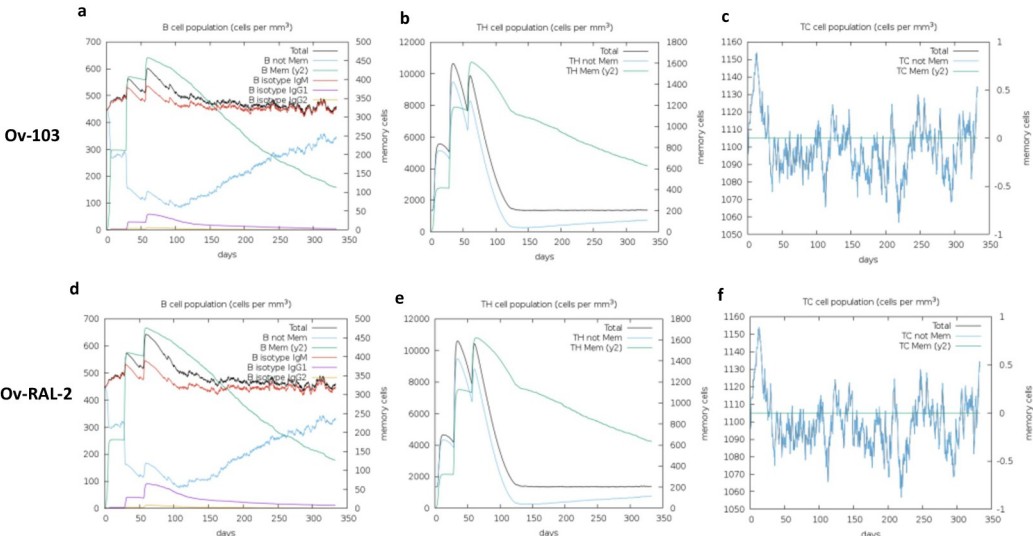

**Fig 3. Immune simulation of B- and T-cell responses for both vaccine candidates.** Upper panels indicate a) B-cell, b) T Helper cell, and c) T cytotoxic cell responses for *Ov*-103; while the lower panels are representations of d) B-cell, e) T Helper cell, and f) T cytotoxic cell responses for *Ov*-RAL-2.

potential to elicit robust and memory-driven immune responses, which are generally essential for sufficient vaccine efficacy.

## Structural modelling and assessment of innate immune recognition of both antigens

**Secondary structure and disorder prediction, tertiary structure modelling, refinement, and validation.** Secondary structure and disorder prediction analysis showed that *Ov*-RAL-2 is comprised of approximately 72% alpha helix, 4.3% beta-strand, and 23.7% coil, while *Ov*-103 is predicted to contain 84.8% alpha helix, 4.4% beta-strand, and 10.8% coil (Fig 4c and 4d). In terms of solvent accessibility, 15% of amino acid residues were predicted to be exposed, 61% medium exposed, and 22% buried. Both vaccine candidates had 292 residues (48%) located in disordered domains according to the RaptorX Property server. The 3D models (Fig 4a and 4b) for both candidates were generated using the RaptorX web server. For *Ov*-RAL-2, the best template used for homology modelling was the crystal structure of an Anisakis simplex allergen (PDB ID: 2mar chain A). The model had a p-value of 9.93e-04, indicating high accuracy modelling [64]. Similarly, the best template used for *Ov*-103 was chain A of the crystal structure of *Klebsiella oxytoca* NasR transcriptional anti-terminator (PDB ID: 4akk chain A). The model also had a significant p-value of 1.09e-03.

To refine the initial models, ModRefiner and GalaxyRefine2 servers were used, resulting in 10 relaxed models for each candidate. The models with the best quality scores including RMSD < 7Å, MolProbity score < 2, clash score < 5, and Ramachandran plot score > 97% were then selected for further analysis. Three models for *Ov*-RAL-2 and 5 models for *Ov*-103 met these thresholds, however, only the top ranked model for each antigen was used for downstream analyses. The selected model for *Ov*-103 exhibited quality assessment scores, including RSMD (6.328Å), MolProbity (0.728), clash score (0.7), poor rotamers (0.0), Ramachandran favoured (99.4%), and GALAXY energy (-4380.64). Meanwhile, for *Ov*-RAL-2, the chosen model scored the following: RSMD (2.843Å), MolProbity (0.797), clash score (1.0), poor

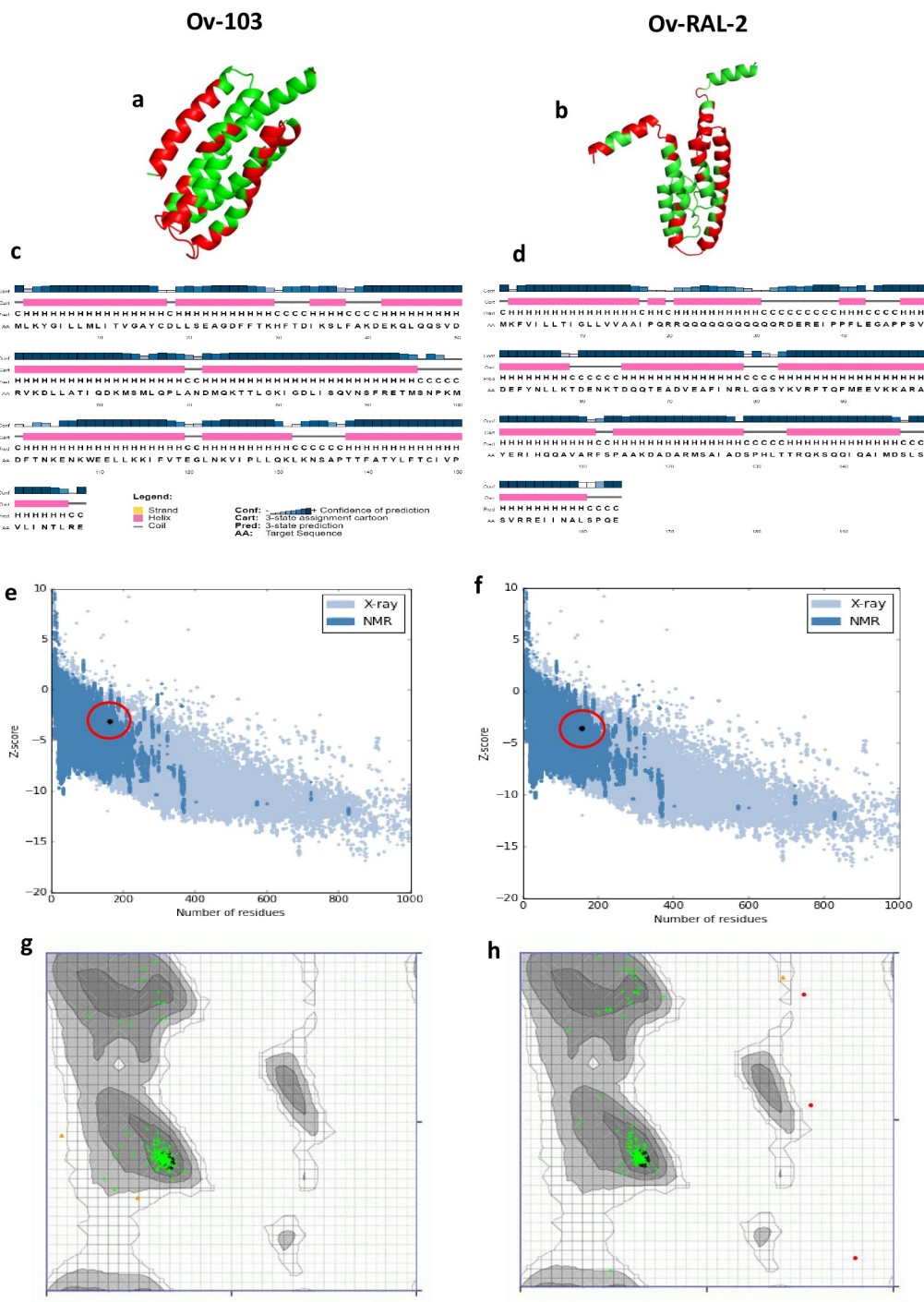

**Fig 4. Structural modelling and quality assessments of vaccine candidates.** a) *Ov*-103 and b) *Ov*-RAL-2 predicted models showing conformational epitopes (coloured green) found in both proteins. c) *Ov*-103 and d) *Ov*-RAL-2 secondary structural features. e) *Ov*-103 and f) *Ov*-RAL-2 graphical presentation of high accuracy predictions relative to experimentally solved structures. g) *Ov*-103 and h) *Ov*-RAL-2 Ramachandran analyses revealing most residues in predicted models localised in favoured regions.

rotamers (0.0), Ramachandran favoured (98.1%), and GALAXY energy (-4341.03). These refined models (Fig 4a and 4b) were selected as the final 3D models for further analysis. The quality and potential errors in the crude 3D models were further verified by ProSA-web, with Z-scores of -3.45 for *Ov*-103 and -3.72 for *Ov*-RAL-2 (Fig 4e and 4f); indicating high accuracy of prediction relative to experimentally solved structures on proteins in the PDB database. Ramachandran plot analysis (Fig 4g and 4h) of the modelled proteins revealed that 98.8% of residues for *Ov*-RAL-2 and 98.1% for *Ov*-103 were in favored regions, consistent with the GalaxyRefine2 analysis. Furthermore, the ERRAT assessment (S1a and S1b Fig) revealed an overall structural quality factor of 98.7% for *Ov*-103 and 94.9% for *Ov*-RAL-2 after refinement, which is crucial in vaccine development due to structural impact of antigens on immunogenicity and efficacy. From a structural perspective, *Ov*-103 appears to have more potential as a vaccine candidate, especially considering its higher alpha helix content.

**Information-driven docking evaluation of antigens' binding affinity for innate immune (TLR-4 receptor) recognition.** We further analysed the potential of both antigens (*Ov*-103 and *Ov*-RAL-2) to interact effectively with the innate immune system via Toll-like Receptor 4 (TLR4), which is known to play a significant role in protective immunity against the larval stages of *O. volvulus* [86]. Studies on TLR4-deficient mice and those with natural TLR4 mutations (C3H/HeJ) have revealed increased susceptibility to various infections including onchocerciasis, due to impaired sensing by the innate immune system [86, 87]. To assess the binding affinities between the TLR4 receptor and each vaccine candidate, information-driven docking was performed. Molecular Docking (virtual screening) is a robust theoretical method which allows for the estimation of binding affinity between molecules virtually [88, 89]. The CPORT server was used to identify active and passive amino acid residues essential for interaction with the TLR-4 receptor, including Phe31, Ile34, Lys35, Phe38, Glu106, Leu156, Arg157 for *Ov*-103; Asp34, Thr64, Asp65, Gln66, Glu69, Val94, Ala97, Arg98, Tyr101 for *Ov*-RAL-2; and Thr35, Ala36, Ala38, Pro39, Val40, Ala41, Val42, Ala43, Ala44, Ala45, Gly46, Ala47, Ala48, Pro49 for the TLR4 agonist. The TLR4 agonist (control antigen) used in this study is a Mycobacterium tuberculosis 50S ribosomal protein (UniProtKB ID P9WHE3) known to have a high affinity for the TLR4 receptor. Additionally, the active residues of TLR4 were identified from an experimentally solved lipopolysaccharide X-ray co-crystallographic model of the receptor, and include Arg264, Lys341, Lys362, Lys388, Asn417, Glu439, Phe440, and Phe463 [68].

The HADDOCK tool requires a pre-defined diving pocket on the surfaces of both ligand and receptor molecules to guide the docking interactions. Therefore, the active residues listed for each antigen above were used as inputs to define these pockets. Each vaccine candidate was then docked towards the active site of TLR4, allowing assessment of the binding affinity for immune recognition. A HADDOCK score, a measure of binding affinity between two molecules used to rank the docked poses during docking, is calculated based on energy terms including Van der Waals energy, Electrostatic energy, Desolvation energy, and Restraints violation energy (Table 3). Once docking calculations were completed, HADDOCK ranked generated vaccine-TLR4 complexes by clusters, resulting in 19 structures in the best-scoring cluster for the *Ov*RAL2-TLR4 complex (representing 41.5% of the water-refined models), 4 structures for *Ov*103-TLR4 complex (accounting for 60.0% of the water-refined models), and 83 structures for the Agonist-TLR4 complex (comprising 69.5% of the water-refined models). These top-ranked clusters for each case with the most negative HADDOCK energy scores for all cases (Table 3) were selected for each vaccine-TLR4 complex and docked-poses manually analysed (Fig 5).

Comparing the HADDOCK scores across the antigens and control agonist groups, the TLR4-agonist complex (positive control) achieved a score of -113.7 +/- 11.2 kcal/mol, while *Ov*-RAL-2 had a score of 106.1 +/- 4.7 kcal/mol, which was higher than that of *Ov*103-TLR4

**Table 3. Energy characterisation of the top-ranked vaccine-TLR4 docked clusters.**

| Parameter* | Protein | | |
|---|---|---|---|
| | TLR4 Agonist | *Ov*-103 | *Ov*-RAL-2 |
| HADDOCK score (kcal/mol) | -113.7 +/- 11.2 | -59.9 +/- 10.5 | -106.1 +/- 4.7 |
| Cluster size with different binding modes to TLR4 | 83 | 4 | 19 |
| RMSD of bound complexes relative to the lowest-energy complex in cluster (Å) | 0.9 +/- 0.6 | 15.0 +/- 0.3 | 24.4 +/- 0.2 |
| Average van der Waals energy (kcal/mol) | -59.5 +/- 7.7 | -27.7 +/- 1.9 | -55.2 +/- 7.2 |
| Average Electrostatic energy (kcal/mol) | -488.3 +/- 20.9 | -417.2 +/- 62.0 | -442.9 +/- 41.9 |
| Average Desolvation energy (kcal/mol) | 35.4 +/- 13.6 | 44.8 +/- 5.6 | 24.6 +/- 6.9 |
| Average Restraints violation energy (kcal/mol) | 81.1 +/- 54.98 | 64.4 +/- 44.74 | 131.0 +/- 21.88 |
| Average Buried Surface volume (Å³) | 2539.3 +/- 270.6 | 1513.1 +/- 86.5 | 2001.7 +/- 50.8 |

*A lower HADDOCK score indicates a strong interaction between proteins. The HADDOCK score calculation is based on van der Waals, Electrostatic, and Desolvation energies.

complex with a score of -59.9 +/- 10.5 kcal/mol. These data may suggest that *Ov*-RAL-2 is potentially more efficient at inducing innate immunity than *Ov*-103. However, the results contrast with the findings from the immune simulation, where *Ov*-103 exhibited higher antibody titers than *Ov*-RAL-2.

**Molecular dynamic simulation suggests *Ov*-103 and *Ov*-RAL-2 form stable complexes with good binding affinities to TLR4.** To further estimate the binding affinity for both antigens onto the TLR4 receptor, molecular dynamic simulation was performed, and the results indicate that *Ov*-103 and *Ov*-RAL-2 form stable complexes with favourable binding affinities to TLR4 (Fig 6). To assess the structural stability of the vaccine candidates, the root mean

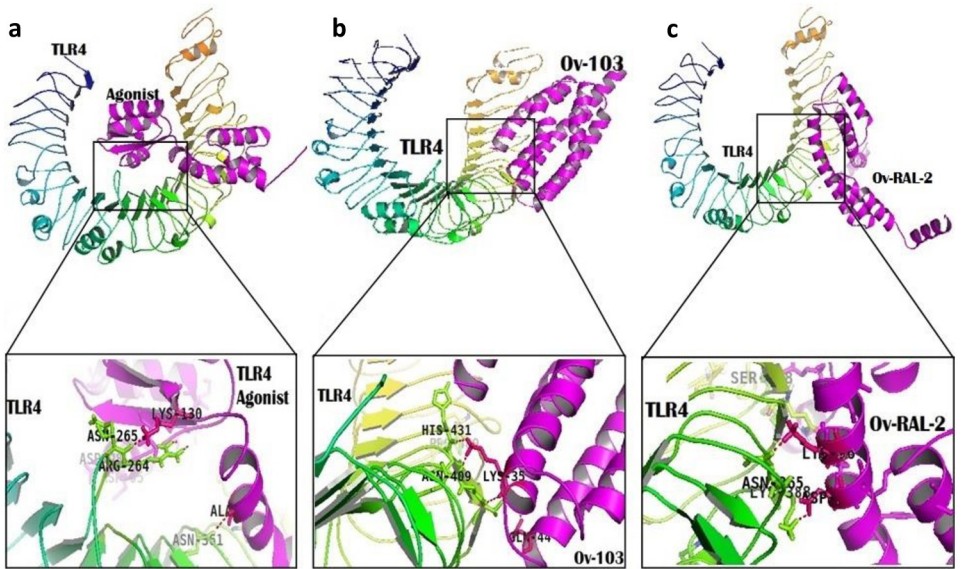

**Fig 5. Docking studies.** Visualisations of the interactions between a) Agonist-TLR4 complex, b) *Ov*103-TLR4 complex, and c) *Ov*RAL2-TLR4 complex showing side chain residues involved in binding.

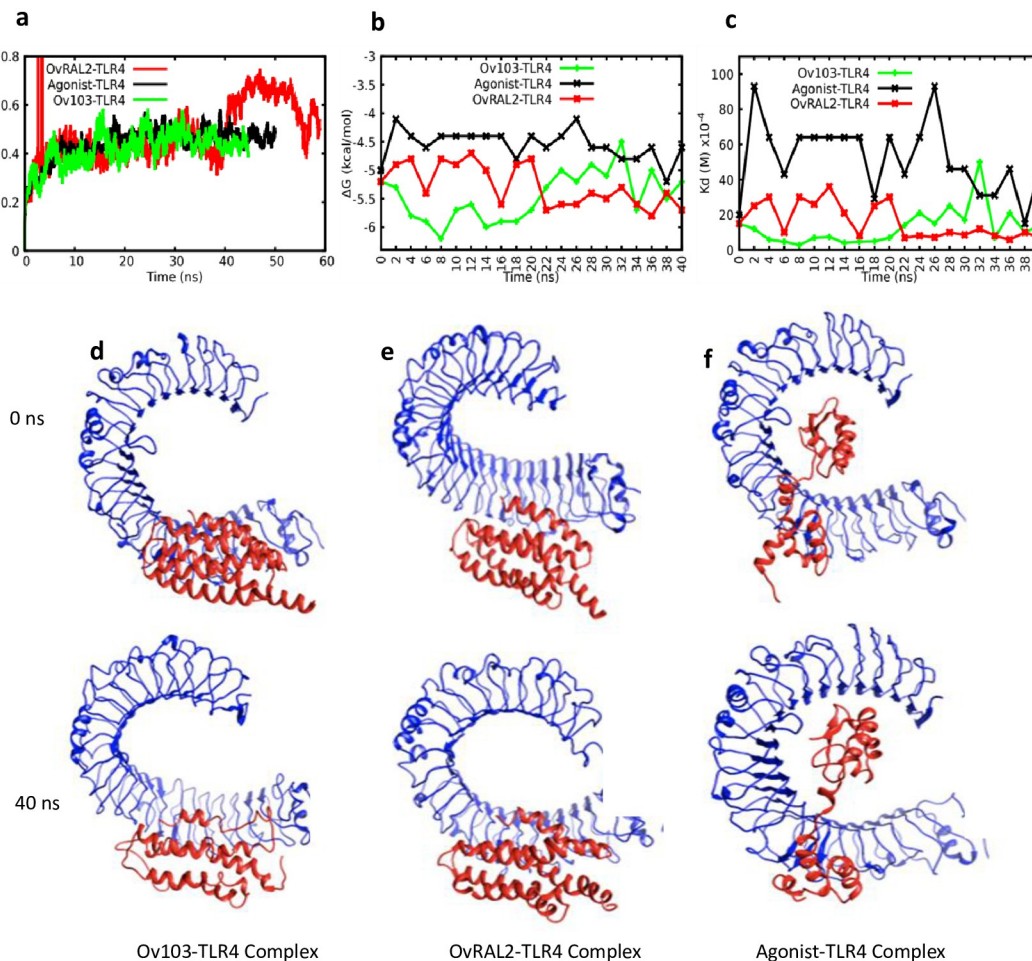

**Fig 6. Molecular dynamics simulation to predict binding affinity and stability of interaction of vaccine candidates over time.** a) Root Mean Square Deviations (RMSD) b) Binding free energy and c) binding dissociation constants of vaccine candidates onto TLR4 innate immune receptor. Structural representation of d) *Ov*103-TLR4 e) *Ov*RAL2-TLR4, and f) Control (Agonist)-TLR4 complexes showing diverse binding conformations over time between 0 ns to 40 ns.

square deviation (RMSD) was analysed and revealed that the *Ov*RAL2-TLR4 complex equilibrated at 10 ns, while the Agonist-TLR4 and *Ov*103-TLR4 complexes equilibrated over 15 ns (Fig 6a). The RMSD value of *Ov*RAL2-TLR4 exhibited fluctuations from 10–40 ns, ranging from 0.3 to 0.5 nm, before rising to 0.6 nm at 55 ns and returning to 0.4 nm. Conversely, the RMSD of Agonist-TLR4 and *Ov*103-TLR4 showed less fluctuation throughout the simulation, with average RMSD values of 0.47 nm and 0.43 nm, respectively. The radius of gyration (Rg), a measure of compactness and global dimension, was also assessed. The Rg indicates that the *Ov*RAL2-TLR4 complex experienced higher changes in Rg, ranging from 3.4 to 3.0 nm between 10 to 47 ns. In contrast, the Agonist-TLR4 and *Ov*103-TLR4 complexes showed less variation in size, with Rg values ranging from 3.1–3.2 nm and 3.2–3.3 nm, respectively (S2a Fig). Both RMSD and Rg suggest that the *Ov*RAL2-TLR4 complex might be less stable compared to the other complexes.

The root mean square fluctuations (RMSF) were also analysed to evaluate the fluctuation of amino acid residues over time. Generally, all RMSF plots showed a greater spread and less correlation. However, for *Ov*RAL2-TLR4, the residues exhibited a distinct behaviour with even

more spread (S2b Fig). This figure shows that *Ov*RAL2-TLR4 has higher residue fluctuations, suggesting that the binding of *Ov*-RAL-2 might be hindered due to these significant changes in residue fluctuations compared to other vaccine constructs. These data are consistent with our previous structural analyses (in the section titled "Secondary structure and disorder prediction, tertiary structure modelling, refinement, and validation") which indicated that *Ov*-103 appears to be more structurally efficient than *Ov*-RAL-2, especially considering that *Ov*-RAL-2 has less alpha helix content. Moreover, the solvent-accessible surface area (SASA), which measures accessibility to solvent molecules with relevance in the study of protein folding and stability, was examined. *Ov*103-TLR4 showed a larger SASA value compared to other complexes (S2c Fig). The probability distribution of SASA values revealed two maxima at 436 and 439 nm$^2$. *Ov*RAL2-TLR4 exhibited a maximum at 429 nm$^2$, followed by the agonist-TLR4 complex, which showed a maximum at 417 nm$^2$ (S2d Fig). These findings suggest that the binding of *Ov*-RAL-2 caused more protein unfolding compared to the other complexes. To evaluate the binding affinity of the vaccine candidates to TLR4, binding free energy and associated dissociation constants were calculated at 2 ns intervals (Fig 6b and 6c). The binding free energy measures the stability of a vaccine construct upon engaging the receptor. Agonist-TLR4 showed less affinity and a high dissociation constant. Surprisingly, *Ov*RAL2-TLR4, which exhibited less affinity during the first 20 ns, had a stronger binding affinity during the last 20 ns. Ultimately, both vaccine constructs exhibited similar binding affinities (-5.2 and -5.7 kcal/mol for *Ov*103-TLR4 and *Ov*RAL2-TLR4, respectively) (Fig 6b and 6c).

## Discussion

Onchocerciasis remains one of the most debilitating neglected tropical diseases (NTDs), leading to severe eye and skin lesions. As the global health approach for combating the disease transitions from control to elimination, there has been a consensus on the need for alternative control tools such as prophylactic/therapeutic vaccines. Several preclinical studies have provided consistent evidence of protective immunity against the disease, targeting various candidate antigens, notably *Ov*-RAL-2 and *Ov*-103 [18–20, 90, 91]. The Onchocerciasis Vaccine for Africa (TOVA) Initiative was established in 2015, aiming to advance at least one vaccine candidate through Phase I human trials by 2025, initially focusing on infants and children below 5 years old [9, 11]. Previous studies explored using both antigens individually, in combination, or as a fused single protein (*Ov*-FUS-1), obtaining promising results in mice and cattle [12, 20, 90, 91]. The fusion protein (*Ov*-FUS-1) formulated with Advax-CpG has demonstrated a more robust protective immune response in both mice and non-human primates, than when *Ov*-RAL-2 and *Ov*-103 are administered separately, by eliciting a balanced Th1/Th2 response that enhances immunogenicity and antibody production, critical for long-term protection against *O. volvulus* [21]. In contrast, co-administration of the two independent proteins resulted in slower larval killing and potentially less effective immune recruitment, highlighting the advantages of a single fusion construct like *Ov*-FUS-1 for efficient immune activation and response durability. However, the overall success rate of vaccine candidates, similar to drugs, during clinical development remains relatively low. In the vaccine development pipeline, bioinformatics tools continue to play crucial roles, facilitating rational antigen design and predictions of efficacy in clinical trials. With clinical trials planned for *Ov*-FUS-1 (the chimera of *Ov*-RAL-2 and *Ov*-103) in 2025 [12, 17], our study focused on predicting the immunogenicity and other important vaccine-related parameters of the independent antigens using immunoinformatics. In the past, a similar approach was deployed to predict that immune escape and immune camouflage could hinder the efficacy of the RTS, S malaria vaccine in Malawi [92].

Certain mechanisms of acquired protective immunity against *O. volvulus* infection in humans have been investigated and revealed that protective immunity is associated with the capacity to generate mixed Th1/Th2 responses against *O. volvulus* L3 and/or molting L3 [15]. Additionally, the presence of cytophilic antibodies, in combination with cytokines produced, facilitates effective anti-L3 antibody-dependent cell-mediated cytotoxicity (ADCC) reactions against L3 [12]. In this study, both vaccine candidates underwent thorough immunoinformatics characterization using multiple tools. Firstly, we assessed various epitopes in the vaccine candidates that represent their potential for stimulating adverse side effects, immunomodulatory (camouflage) properties, antibody-mediated (humoral) responses, cellular immune responses, and cytokine-based responses which mediate between the humoral and cellular responses. This was done using different modelling tools with distinct parameters and training datasets. Secondly, the study evaluated the ability of each vaccine candidate to be recognized by the innate immune TLR-4 receptor, using structural biology methods. Finally, we performed a virtual vaccination simulation to model the primary and secondary humoral, cellular, and cytokine-mediated responses over time.

The analysis revealed that these candidates were safe, antigenic, non-camouflageable, and contained a diverse array of epitope types known to be associated with onchocerciasis protective response. These epitopes include B-cell epitopes (both linear and conformational) and T-cell epitopes (CD4+ and CD8+) with specificities for different HLA alleles, and various relevant cytokine epitopes, including IFN-γ, IL-4, IL-5, IL-17, IL-10, and TNF-α. The presence of many predicted epitopes for different immune components provides support for the experimentally determined immune correlates of protection against the infection [15, 16]. Additionally, the data aligns with our *in silico* immune simulation analysis, which forecasts an increase in antibody production after immunization, along with elevated levels of IFN-γ secretion and a sustained cellular response for both antigens. The analyses also anticipate a rise in antibody levels for both antigens following the booster doses, indicating the development of immune memory—a crucial aspect of vaccination. We also evaluated the potential of the antigens to stimulate the innate immune TLR4 receptor, and the findings suggest that they both interact with the receptor with favourable binding affinities, implying potential TLR4 immune recognition.

Both *Ov*-RAL-2 and *Ov*-103 exhibit favourable safety profiles, as they were predicted to have no toxic signals and were identified as non-allergenic. Onchocerciasis vaccination targets the prevention of infection in children under 5 years old, for a start. A study in a highly endemic region in Ghana tested anti-*Ov*-103 and anti-*Ov*-RAL-2 IgE responses in infants and children aged 1–5 years vs. 6–8 years [79], respectively. None of the participants under 5 had elevated functional IgE responses. This suggests that continuous exposure to infective larvae with native *Ov*-103 and *Ov*-RAL-2 proteins did not induce functional IgE in young children, reducing concerns about pathological atopic responses in vaccinated children [12]. Taken together, these findings suggest that both proteins would very likely demonstrate a favourable safety profile in humans. Cytokine analyses revealed that *Ov*-103 contains a significant number of IL-17 epitopes (82 epitopes). IL-17 is commonly associated with promoting inflammation and contributing to the development of allergic reactions and autoimmune diseases. Interestingly, both *Ov*-103 and *Ov*-RAL-2 antigens were also found to harbour substantial amounts of IL-10 epitopes (82 and 87 epitopes, respectively). IL-10 is known to counteract excessive immune responses and inflammation, playing a critical role in maintaining immune homeostasis and preventing autoimmune diseases and local toxicity. However, it's important to note that IL-10's presence could potentially limit the parasite-killing capability during vaccination, due to its immunomodulatory effects [93].

In addition, both candidates were also predicted to contain substantial numbers of IL-5-inducing epitopes and TNF-α-inducing epitopes. Studies done on a Cameroonian population reported more than 80% of PBMC samples obtained from putatively immune individuals produced IL-5 in response to the infective L3 antigens [16]. Meanwhile, another study focused on assessing the immune responses in a Ghanaian population concluded that mf negativity is associated with IL-5 and IL-13 production [94]. With regards to TNF-α, it has been reported that *O. volvulus* adult worm extracts obtained from untreated patients showed neutrophil chemotactic activity and induced strong TNF-α and IL-8 production in human monocytes [95]. Monocytes have been implicated in the inhibition of L3 larval molting in the presence of either *Ov*-103 antibodies or *Ov*-RAL-2 antibodies [18]. The results obtained from these different studies suggest the important roles of both TNF-α and IL-5 in the generation of protective immunity against onchocerciasis.

*Ov*-RAL-2 and *Ov*-103 were assessed for cross-conservation with the human proteome. While a few T-cell epitopes in *Ov*-RAL-2 matched peptides in some human proteins, *Ov*-103 showed no homology with the human proteome, making it potentially non-camouflageable. However, *Ov*-RAL-2 contained a low complexity Glutamine Repeat Region (GRR) with sequence identity to similar GRR regions in different human proteins, which could be a potential mechanism to contribute to sub-optimal protection against onchocerciasis, although the GRR region of *Ov*-RAL-2 accounts for only 6.7% of the full protein's length, thus, may result in only negligible tolerogenicity. It may, however, serve as a potential epitope that provides leverage for regulatory T cells (Tregs) to induce immune escape [92] and elicit partial immune reactions to the antigen, contributing to sub-optimal protection against onchocerciasis. This antigen could thus be optimised by modifying or trimming off the GRR region to improve the chances of avoiding potential immune modulation in humans. *In vitro* and animal studies testing a hypothesis about the impact of the GRR region on *Ov*-RAL-2 efficacy would greatly benefit the advancement of this antigen to a vaccine.

Both *Ov*-RAL-2 and *Ov*-103 exhibited significant antigenic properties with *Ov*-103 obtaining slightly higher scores, suggesting it may be slightly more antigenic. Both antigens were predicted to have numerous B-cell epitopes, indicating their potential to generate potent antibody responses. The number of B-cell epitopes was higher for *Ov*-RAL-2, suggesting it could be more antigenic in terms of B-cell response. Both *Ov*-RAL-2 and *Ov*-103 contain CTL and HTL epitopes, indicating their potential to be recognized by MHC class I and II molecules, eliciting appropriate immune responses, although *Ov*-RAL-2 had more predicted CTL epitopes, while *Ov*-103 had more predicted HTL epitopes. Overall, it appears that *Ov*-103 has some advantages over *Ov*-RAL-2 in terms of antigenicity and safety, as it had no homology with human proteins, no GRR region, and slightly higher antigenicity scores. *Ov*-103 exhibits a more robust protective immune response compared to *Ov*-RAL-2, as it contains a higher number of Ig-inducing epitopes, capable of promoting both IgA and IgG production, indicating its potential to elicit a broader and stronger humoral immune response. The immune simulation results also suggest that *Ov*-103 induces a more sustained primary immune response and a more specific and long-lasting secondary and tertiary immune response, as reflected by higher and sustained levels of IgM and IgG1+IgG2 antibodies. Both antigens show promise in triggering a mix of Th1 and Th2 responses, which is essential for targeting onchocerciasis through vaccination.

Information-driven docking evaluation for TLR-4 recognition revealed that *Ov*-RAL-2 binds with greater affinity to the TLR4 receptor compared to *Ov*-103, suggesting a potentially better affinity for inducing innate immunity. However, this contrasts with the immune simulation findings, where *Ov*-103 induced higher antibody titres than *Ov*-RAL-2, implying a discrepancy between binding affinity to TLR-4 receptor and immune response efficacy. During

the molecular dynamics simulation analyses, both *Ov*-103 and *Ov*-RAL-2 formed stable complexes with good binding affinities to TLR4 over time. *Ov*-103 displayed more structural efficiency and a larger solvent-accessible surface area, contributing to its structural stability as a peptide vaccine. In contrast, *Ov*-RAL-2 exhibited more fluctuations in RMSD and Rg, indicating potential instability in the complex. Interestingly, these findings align with previous physicochemical analyses of both antigens, wherein the instability index (II) of *Ov*-103 was 37.44, while *Ov*-RAL-2 exhibited a value of 67.79. An instability index (II) exceeding 40 indicates instability, thus suggesting that *Ov*-RAL-2 displayed an unstable nature.

## Challenges and limitations

This study mainly relies on *in silico* computational methods to assess the immunogenicity and safety of vaccine candidates. While these methods provide valuable insights, they lack direct experimental validation in real biological systems. Some challenges include:

1. the absence of *in vivo* studies and animal testing which might limit the translation of the findings to practical vaccine development, however, the data generated herein provides valuable insights pointing researchers to potential angles for optimisation and trial design;

2. the contrasting results between immune simulation and binding affinity evaluations raise questions about the relationship between binding affinity and actual immune response efficacy. This limitation could result from;

3. the fact that this study primarily evaluates the binding affinity of vaccine candidates to TLR4. However, the immune response involves complex interactions between multiple receptors and pathways. A thorough assessment of the vaccine candidates interactions with additional immune receptors, such as Toll-like receptor-2 (TLR2), nucleotide-binding domain, leucine-rich repeat-containing (NLR) receptors, C-type lectin receptors (CLRs), and Dectin-1, which are all directly involved in the innate immune recognition of helminth parasites, including *O. volvulus* [96], could offer a more comprehensive understanding of their efficacy;

4. the study also uses homology modelling based on template structures, which might not perfectly represent the actual 3D structures of the vaccine candidates. This limitation could have affected the accuracy of the predictions, potentially including a certain degree of uncertainty in our conclusions;

5. the study focuses on the binding affinity between the vaccine candidates and TLR4 but does not consider the subsequent antigen processing and presentation which are critical for the generation of an effective immune response. Therefore, it is important to indicate that these findings should be interpreted with caution.

## Conclusion

In conclusion, despite the challenges and limitations of our approach, *in silico* tools as used herein could provide powerful insights in the design of targeted clinical trials by forecasting immune responses to candidate antigens. Therefore, this study provides several layers of evidence supporting the safety and vaccine potentials of *Ov*-103 and *Ov*-RAL-2 for human protection against onchocerciasis, as well as strongly advocates for the advancement of these antigens through clinical development to help meet the high demand for urgent onchocerciasis vaccines.

## Supporting information

**S1 Fig.** Overall template-based structural modeling quality factor of a) *Ov*-103 and b) *Ov*-RAL-2.
(TIF)

**S2 Fig.** Molecular dynamics simulation evaluations of a) Radius of gyration (compactness) of antigens bound to TLR4 receptor b) Root mean square fluctuation (RMSF) c) Solvent accessible surface area (SASA) and d) Probabilities of SASA deviations.
(TIF)

**S1 File. *Ov*-103 predicted epitopes.** Tables 1–10 classifying distinct epitope types identified in *Ov*-103.
(XLSX)

**S2 File. *Ov*-RAL-2 predicted epitopes.** Tables 1–10 classifying distinct epitope types identified in *Ov*-RAL-2.
(XLSX)

## Acknowledgments

We thank Dr. Nkemngo Francis Nongley and Dr. Nelly Manuela Tatchou for providing valuable insights and Dr. Kum Kevin Esoh for generating the immune simulation data.

## Author Contributions

**Conceptualization:** Robert Adamu Shey, Fidele Ntie-Kang, Stephen Mbigha Ghogomu.

**Data curation:** Derrick Neba Nebangwa, Robert Adamu Shey, Daniel Madulu Shadrack, Cabirou Mounchili Shintouo, Ntang Emmaculate Yaah, Bernis Neneyoh Yengo, Mary Teke Efeti, Ketura Yaje Gwei, Darling Bih Aubierge Fomekong, Gordon Takop Nchanji, Arnaud Azonpi Lemoge.

**Formal analysis:** Derrick Neba Nebangwa, Robert Adamu Shey, Daniel Madulu Shadrack.

**Investigation:** Derrick Neba Nebangwa, Robert Adamu Shey, Daniel Madulu Shadrack, Cabirou Mounchili Shintouo, Mary Teke Efeti, Arnaud Azonpi Lemoge.

**Methodology:** Derrick Neba Nebangwa, Robert Adamu Shey, Daniel Madulu Shadrack.

**Project administration:** Fidele Ntie-Kang, Stephen Mbigha Ghogomu.

**Resources:** Derrick Neba Nebangwa, Robert Adamu Shey, Daniel Madulu Shadrack, Fidele Ntie-Kang, Stephen Mbigha Ghogomu.

**Software:** Derrick Neba Nebangwa, Daniel Madulu Shadrack.

**Supervision:** Robert Adamu Shey, Fidele Ntie-Kang, Stephen Mbigha Ghogomu.

**Visualization:** Derrick Neba Nebangwa.

**Writing – original draft:** Derrick Neba Nebangwa, Robert Adamu Shey, Daniel Madulu Shadrack.

**Writing – review & editing:** Derrick Neba Nebangwa, Robert Adamu Shey, Daniel Madulu Shadrack, Fidele Ntie-Kang, Stephen Mbigha Ghogomu.

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
