## [Decision Letter · Decision Letter 0]

14 Aug 2024

PONE-D-24-20348Predictive Immunoinformatics Reveal Promising Safety and Anti-Onchocerciasis Protective Immune Response Profiles to Vaccine Candidates (Ov-RAL-2 and Ov-103) in Anticipation of Phase I Clinical TrialsPLOS ONE

Dear Dr. Nebangwa,

Thank you for submitting your manuscript to PLOS ONE. After careful consideration, we feel that it has merit but does not fully meet PLOS ONE’s publication criteria as it currently stands. Therefore, we invite you to submit a revised version of the manuscript that addresses the points raised during the review process.

We look forward to receiving your revised manuscript.

Kind regards,

Anoop Kumar, Ph.D.

Academic Editor

PLOS ONE

Journal Requirements:

Reviewers' comments:

Reviewer's Responses to Questions

**Comments to the Author**

1. Is the manuscript technically sound, and do the data support the conclusions?

Reviewer #1: Partly

Reviewer #2: Yes

Reviewer #3: Yes

2. Has the statistical analysis been performed appropriately and rigorously? 

Reviewer #1: I Don't Know

Reviewer #2: N/A

Reviewer #3: Yes

3. Have the authors made all data underlying the findings in their manuscript fully available?

Reviewer #1: Yes

Reviewer #2: Yes

Reviewer #3: Yes

4. Is the manuscript presented in an intelligible fashion and written in standard English?

Reviewer #1: Yes

Reviewer #2: Yes

Reviewer #3: Yes

5. Review Comments to the Author

Reviewer #1: The authors have evaluated various in silico tools to identify potential targets for onchocerciasis, somewhat thoroughly. While they have employed varied tools, the manuscript is lacking coherence in flow of topics/ ideas/ tools. Also, so many different thresholds have been employed by the authors, varying based on the tool/ epitope evaluated while no information or background was provided about the reason for such differences. I highly compel the authors to address this lack of a global standard, and enhance the reach of the manuscript by discussing about how one epitope prediction necessitates / flows into another.

Reviewer #2: The study adds to the existing evidence of safe and protective immune response generation by the Ov-RAL-2 and Ov-103 vaccine candidates for human protection against onchocerciasis. This immunoinformatics based profiling of these two potential vaccine candidates provides significant information in anticipation of phase I clinical trials.

Comments:

1. Antigenicity of both Ov-RAL-2 and Ov-103 were not found significant when tested by ANTIGENpro, and antigenicity score of both antigens by VaxiJen was also found on the borderline of significance cutoff (0.5). Can the authors support this data with some evidence from literature where significance estimated by these tools is lesser but the antigens showed good response (antigenicity) in clinical trials. The authors can also use an additional tool to find if antigenicity is significant or not.

2. While this study is focused to characterize the profiles of each candidate individually, it will add to the study if the authors can also comment about profiling for multi-epitope vaccine design including the epitopes from both the antigens in the discussion, and cite the relevant study from existing literature about co-administration of the two proteins for generation of durable protective immune response against O. volvulus (PMID: 37515028).

3. One of the strong challenges identified from this study is the observed contrast between immune simulation and binding affinity and the authors have well stated that a more comprehensive evaluation of the vaccine candidate interactions with other immune receptors in addition to TLR4 can provide a more complete picture of their efficacy. The authors should mention here some of the other immune receptors (or receptor complex proteins) from the literature that should be tested in future for their binding efficacy to the antigens.

4. I suggest the authors to move plot legends (for example Fig 2c and 2d) in the empty space (below the inset plots) so that both plots and legends are clearer.

5. Figure 5, c) is missing in legend

6. References 82 and 86 are same. Authors should re-check all the references and their respective citations in the manuscript before resubmission.

Reviewer #3: Overall, the manuscript titled, “Predictive Immunoinformatics Reveal Promising Safety and Anti-Onchocerciasis Protective Immune Response Profiles to Vaccine Candidates (Ov-RAL-2 and Ov-103) in Anticipation of Phase I Clinical Trials” is well written, with in -depth computational analysis performed to predict the immune responses generated in humans against the nematode upon host infection. The data is sufficient supporting the conclusions and the authors acknowledge the challenges during the study design in terms of lack of experimental data, contrasting results between immune simulation and binding affinity evaluation, limitation to predict accurate 3D vaccine candidates’ structure etc. Introduction contains sufficient background information regarding the tropical onchocerciasis disease pathogenesis and control measures with few edits required as mentioned in comments below. I believe the manuscript requires few edits and can be accepted for the publication.

Line 60 – The global 60 incidence is estimated to reach approximately 20.9 million infections. Please indicate the year in this sentence. By which year is this number of infections estimated?

Line 65 – needs re-framing the sentence.

Line 68 – Explain what is CDTi based approach for elimination of disease. Many readers will not be aware of it.

Line 76 – This sentence needs more explanation. It is not necessary that every vaccine developed for an infectious disease, or any other tropical disease would not generate drug resistance to parasite. Drug resistance bacteria for example against Tuberculosis develops despite the clinically approved drugs in market like Bedaquiline.

Line 103 – Refer the research papers examining immune responses developed to onchocerciasis in both human and animal models.

Line 197 and 422 – Paragraph, “Cytokine epitope prediction for IFN-γ, IL-4, IL-17, and IL-10”. Were epitopes for other pro-inflammatory cytokines CD154, TNF-a, IL-2 induced in infectious diseases also predicted? It will be relevant to predict TNF-a epitopes in humans considering it is a major proinflammatory cytokine released by macrophages.

Line 231 – Paragraph, “Ig-class prediction”. I was curious about the role of IgM antibody secretion by B cells during the pathogenesis of the nematode? Were epitopes predicted for IgM antibody along with other antibodies IgG, IgA and IgE studied? In viral infections IgM is known to be released during acute timepoints (3 to 4 days) post infection but its level is not sustained for a longer duration or is not detected.

Line 654 – “With clinical trials planned for Ov-Fus-1 (a chimera of Ov-RAL-2 and Ov-103) in 2025 [12, 16], our study focused on predicting the immunogenicity and other important vaccine-related parameters of both antigens using immunoinformatics”. Are there any other prophylactic or therapeutic vaccine candidates currently in clinical trials for elimination of onchocerciasis disease? If yes, this should be mentioned in discussion alongside above statement.

6. PLOS authors have the option to publish the peer review history of their article (what does this mean?). If published, this will include your full peer review and any attached files.

Reviewer #1: No

Reviewer #2: No

Reviewer #3: **Yes: **Rakhi Harne

---

## [Author Response · Author response to Decision Letter 0]

18 Sep 2024

The Editor,

PLOS ONE Journal

Buea, 18th September 2024

Dear Editor,

Rebuttal: PLOS ONE Decision: Revision required [PONE-D-24-20348] - [EMID:bb24 646e0 dd9b439]

We sincerely appreciate your thorough and insightful evaluation of our study. The editor’s and reviewers’ comments have significantly enhanced our work and broadened our understanding of how the scientific community perceives our research. Following PLOS ONE’s guidelines, we have addressed all review comments in the revised manuscript and noted them in this letter for your reference. Please, kindly see authors’ feedback for each comment in blue text below:

Reviewer #1: “The authors have evaluated various in silico tools to identify potential targets for onchocerciasis, somewhat thoroughly. While they have employed varied tools, the manuscript is lacking coherence in flow of topics/ ideas/ tools. Also, so many different thresholds have been employed by the authors, varying based on the tool/ epitope evaluated while no information or background was provided about the reason for such differences. I highly compel the authors to address this lack of a global standard, and enhance the reach of the manuscript by discussing how one epitope prediction necessitates/flows into another.”

Coherence in flow of topics and Ideas:

We agree with the reviewer's feedback that the coherence and flow of topics and ideas in the manuscript could be improved. In response, we have made significant revisions to the methods, results, and discussion sections of the manuscript.

Specifically, we have improved the coherence and flow by:

1. Combining several standalone topics and grouping them under newly introduced major headings related to different immune components in both the methods and results sections.

2. Presenting the previously standalone topics as paragraphs under these new major headings, rather than as separate subheadings.

3. Adding a high-level "Research design" section in the methods and results to provide an overarching description of the study approach (around lines 202 (methods section) and 637 (result section) of the revised manuscript with tracked changes).

4. Finally, incorporating a research design statement in the discussion section (around line 1157 of the revised manuscript with tracked changes) to further contextualise the study in that segment.

It is our hope that these revisions, which are visible through “Tracked Changes” in the re-submitted manuscript, have significantly improved the clarity and organisation of the paper, and illustrates in a big picture fashion why each epitope class was predicted and how one epitope prediction necessitates / flows into another.

Global standard threshold of tools used:

We acknowledge the reviewer's observation regarding the use of various threshold values across different tools in our study. It's important to note that our research employed more than a dozen modeling tools, each with its unique set of DEFAULT parameters. Furthermore, as described in the relevant sections, these tools were individually trained on distinct experimental datasets. The newly added research design sections in the methods, results, and discussion sections of the paper include new information explaining the rationale for the different thresholds used as explained above.

Given this diversity, applying a single, universal threshold value to assess our predictions isn't quite practical. Instead, we have used tool-specific default thresholds, as detailed in the relevant sections of the paper. These individual thresholds have proven effective in identifying promising epitopes for each topic discussed in the manuscript.

Nevertheless, we remain receptive to any specific recommendations the Review Team might offer regarding the establishment of a standardised global threshold, should they deem it feasible.

Reviewer #2: The study adds to the existing evidence of safe and protective immune response generation by the Ov-RAL-2 and Ov-103 vaccine candidates for human protection against onchocerciasis. This immunoinformatics based profiling of these two potential vaccine candidates provides significant information in anticipation of phase I clinical trials.

Comments:

1. Antigenicity of both Ov-RAL-2 and Ov-103 were not found significant when tested by ANTIGENpro, and the antigenicity score of both antigens by VaxiJen was also found on the borderline of significance cutoff (0.5). Can the authors support this data with some evidence from literature where significance estimated by these tools is lesser but the antigens showed good response (antigenicity) in clinical trials. The authors can also use an additional tool to find if antigenicity is significant or not.

We are grateful to the reviewer for this crucial critique in pointing out the discrepancy in the threshold values used for the antigenicity predictors. Although we initially reported a threshold score of 0.937603 for the ANTIGENpro server, similar to the Vaxijen servers, the actual significant threshold cutoff is 0.5 We have now made this correction in the manuscript (around line 694 of the revised manuscript with tracked changes) and have provided appropriate citations to this cutoff for both tools. Additionally, we have now supported our data with experimental studies demonstrating high levels of antigenicity for both antigens in mice and human studies. We have also demonstrated that other studies using both computational and pre-clinical methods found that an antigen, initially predicted to have low to moderate in silico antigenicity score, later elicited significantly high humoral (antigenicity) responses in mice. These revisions have been made in the first paragraph titled “Prediction of antigenicity of Ov-RAL-2 and Ov-103 in humans:”, in the “Assessment of Humoral Epitope Content of Vaccine Candidates” result section of the revised manuscript.

2. While this study is focused to characterize the profiles of each candidate individually, it will add to the study if the authors can also comment about profiling for multi-epitope vaccine design including the epitopes from both the antigens in the discussion, and cite the relevant study from existing literature about co-administration of the two proteins for generation of durable protective immune response against O. volvulus (PMID: 37515028).

We thank the reviewer for this important suggestion, and we agree that including additional details about the fusion protein (Ov-FUS-1) profile enhances the value of this work. Consequently, we have added two sentences in the first paragraph of the discussion section (around line 1203 of the revised manuscript with tracked changes) to describe the superior protective immune profile of Ov-FUS-1 in both mice and non-human primates compared to the independent Ov-103 and Ov-RAL-2 profiles, and citing this as “reference 21” (PMID: 37515028)

3. One of the strong challenges identified from this study is the observed contrast between immune simulation and binding affinity and the authors have well stated that a more comprehensive evaluation of the vaccine candidate interactions with other immune receptors in addition to TLR4 can provide a more complete picture of their efficacy. The authors should mention here some of the other immune receptors (or receptor complex proteins) from the literature that should be tested in future for their binding efficacy to the antigens.

We appreciate the reviewer’s valuable suggestion. In response, we have now incorporated key innate immune receptors involved in the recognition of O. volvulus into point 3 of the “Challenges and Limitations” section (around line 1358) of the revised manuscript with tracked changes. These receptors include Toll-like receptor-2 (TLR2), nucleotide-binding domain, leucine-rich repeat-containing (NLR) receptors, C-type lectin receptors (CLRs), and Dectin-1, all of which play a role in the innate immune recognition of helminth parasites, including Onchocerca volvulus. The relevant citation has also been added.

4. I suggest the authors move plot legends (for example Fig 2c and 2d) in the empty space (below the inset plots) so that both plots and legends are clearer.

We appreciate the reviewer for this suggestion. We think we have now addressed this, and remain receptive to further guidance on how to do this, if not yet corrected.

5. Figure 5, c) is missing in legend

We have now added the 5c label in the legend.

6. References 82 and 86 are the same . Authors should re-check all the references and their respective citations in the manuscript before resubmission.

We have corrected the duplicate entries of the same reference in our EndNote database, which has resolved the issue. In addition, we have checked for other duplicated entries and ensured that the in-text citations match the respective references in the bibliographic list.

Reviewer #3: Overall, the manuscript titled, “Predictive Immunoinformatics Reveal Promising Safety and Anti-Onchocerciasis Protective Immune Response Profiles to Vaccine Candidates (Ov-RAL-2 and Ov-103) in Anticipation of Phase I Clinical Trials” is well written, with in -depth computational analysis performed to predict the immune responses generated in humans against the nematode upon host infection. The data is sufficient supporting the conclusions and the authors acknowledge the challenges during the study design in terms of lack of experimental data, contrasting results between immune simulation and binding affinity evaluation, limitation to predict accurate 3D vaccine candidates’ structure etc. Introduction contains sufficient background information regarding the tropical onchocerciasis disease pathogenesis and control measures with few edits required as mentioned in comments below. I believe the manuscript requires few edits and can be accepted for publication .

These comments from the reviewer are well appreciated. And we have addressed their concerns in the relevant sections below.

Line 60 – The global 60 incidence is estimated to reach approximately 20.9 million infections. Please indicate the year in this sentence. By which year is this number of infections estimated?

We have now added the year “2017” for which this statistic was taken around line 135 in the revised manuscript with tracked changes.

Line 65 – needs re-framing the sentence.

This has now been re-framed around line 137 in the revised manuscript with tracked changes.

Line 68 – Explain what is a CDTi based approach for elimination of disease. Many readers will not be aware of it.

Many thanks to the reviewer for this important suggestion. We have now added more explanation about the CDTi control programme at around around line 142 in the revised manuscript with tracked changes.

Line 76 – This sentence needs more explanation. It is not necessary that every vaccine developed for an infectious disease, or any other tropical disease would not generate drug resistance to parasites. Drug resistance bacteria for example against Tuberculosis develops despite the clinically approved drugs in the market like Bedaquiline.

We agree with the reviewer’s suggestion for additional explanation, but we also feel the sentence is sufficiently clear on its own. Our aim in writing the background of the manuscript was to maintain a focus on predictive immunoinformatics of the antigens’ profiles without delving too deeply into other general topics like drug discovery.

Line 103 – Refer to the research papers examining immune responses developed to onchocerciasis in both human and animal models.

Many thanks to the reviewer for this observation. We have now cited the relevant studies as references 18 – 21 around line 184 of the revised manuscript with tracked changes.

Line 197 and 422 – Paragraph, “Cytokine epitope prediction for IFN-γ, IL-4, IL-17, and IL-10”. Were epitopes for other pro-inflammatory cytokines CD154, TNF-a, IL-2 induced in infectious diseases also predicted? It will be relevant to predict TNF-a epitopes in humans considering it is a major proinflammatory cytokine released by macrophages.

While the roles of several cytokines, in addition to those analyzed in the manuscript (including TNF-alpha and IL-5) have been reported in the development of protective and/or pathogenic immune responses against onchocerciasis, at the time most of the analyses reported in this paper, only the servers selected for the chosen cytokines in his paper were available. However, we want to note that servers are currently available for the prediction of epitopes capable of inducing IL-5 and TNF-α, and the analyses of antigens for epitopes inducing these responses have been included in the manuscript. First, in the methods section at line 377 for IL-5 and line 413 for TNF-α. Second, in the results section at around line 805 (IL-5) and 818 (TNF-α) in the revised manuscript with tracked changes. These data have been appropriately cited in-text and added to the supplementary files (S1 and S2 Files).

Line 231 – Paragraph, “Ig-class prediction”. I was curious about the role of IgM antibody secretion by B cells during the pathogenesis of the nematode? Were epitopes predicted for IgM antibody along with other antibodies IgG, IgA, and IgE studied? In viral infections, IgM is known to be released during acute time points (3 to 4 days) post-infection but its level is not sustained for a longer duration or is not detected.

We appreciate the reviewer for this comment. The IgPred server, which was used in this study, predicts antibody induction only for IgG, IgA, and IgE. However, we would like to indicate that different studies have reported elevated levels of IgM (in addition to IgG, IgA, and IgE) in humans infected with O. volvulus. In addition, sera derived from bovine calves exposed to O. ochengi (the cattle model for O. volvulus) transmission in a hyperendemic area have also been reported to contain IgM and IgG1 antibodies specific for O. volvulus-derived recombinant proteins.

Line 654 – “With clinical trials planned for Ov-Fus-1 (a chimera of Ov-RAL-2 and Ov-103) in 2025 [12, 16], our study focused on predicting the immunogenicity and other important vaccine-related parameters of both antigens using immunoinformatics”. Are there any other prophylactic or therapeutic vaccine candidates currently in clinical trials for elimination of onchocerciasis disease? If yes, this should be mentioned in discussion alongside the above statement.

We appreciate this comment and question from the reviewer. We like to indicate that the current reality with onchocerciasis is that, being a neglected tropical disease, this is the first clinical trial ever for the onchocerciasis vaccine. There are currently no vaccine trials ongoing for onchocerciasis.

Revisions made in addition to Reviewers’ suggestions 

1. Updated abstract included to replace the former one

2. Reviewer #1 comments written in red on the “PONE-D-24-20348 review.pdf” document have been addressed as Reviewer-Author dialogue, at the appropriate manuscript sections, as can be seen in the revised manuscript with Tracked changes.

3. The entire manuscript has been run through the grammarly software to check any further grammatical errors and be corrected.

Additional Journal Requirements:

Authors’ Feedback: The manuscript has now been revised to meet these standards

2. Please note that PLOS ONE has specific guidelines on code sharing for submissions in which author-generated code underpins the findings in the manuscript. In these cases, all author-generated code must be made available without restrictions upon publication of the work. Please review our guidelines at https://journals.plos.org/plosone/s/materials-and-software-sharing#loc-sharing-code and ensure that your code is shared in a 

---

## [Decision Letter · Decision Letter 1]

4 Oct 2024

Predictive immunoinformatics reveal promising safety and anti-onchocerciasis protective immune response profiles to vaccine candidates (Ov-RAL-2 and Ov-103) in anticipation of phase I clinical trials

PONE-D-24-20348R1

Dear Dr. Derrick Neba Nebangwa,

We’re pleased to inform you that your manuscript has been judged scientifically suitable for publication and will be formally accepted for publication once it meets all outstanding technical requirements.

Kind regards,

Anoop Kumar, Ph.D.

Academic Editor

PLOS ONE

Additional Editor Comments (optional):

Reviewers' comments:

Reviewer's Responses to Questions

**Comments to the Author**

1. If the authors have adequately addressed your comments raised in a previous round of review and you feel that this manuscript is now acceptable for publication, you may indicate that here to bypass the “Comments to the Author” section, enter your conflict of interest statement in the “Confidential to Editor” section, and submit your "Accept" recommendation.

Reviewer #1: All comments have been addressed

Reviewer #2: All comments have been addressed

Reviewer #3: All comments have been addressed

2. Is the manuscript technically sound, and do the data support the conclusions?

Reviewer #1: Yes

Reviewer #2: Yes

Reviewer #3: Yes

3. Has the statistical analysis been performed appropriately and rigorously? 

Reviewer #1: Yes

Reviewer #2: Yes

Reviewer #3: Yes

4. Have the authors made all data underlying the findings in their manuscript fully available?

Reviewer #1: (No Response)

Reviewer #2: Yes

Reviewer #3: Yes

5. Is the manuscript presented in an intelligible fashion and written in standard English?

Reviewer #1: Yes

Reviewer #2: Yes

Reviewer #3: Yes

6. Review Comments to the Author

Reviewer #1: The authors have refined the manuscript to a great extent that has improved the readability and scientific flow of ideas. The manuscript is now technically sound and intelligible.

Please find attached reviewer document containing couple minor suggestions. Also, the authors need to ensure correct formatting since the headings seem to be sized differently, to make it publication ready.

Reviewer #2: The authors have addressed all my comments for their original submission. Additionally, I find that the revised manuscript shows significant improvement over the original, with enhanced grammar and better overall coherence.

Reviewer #3: Thank you to the Authors. All my revision comments and queries have been addressed in the manuscript.

7. PLOS authors have the option to publish the peer review history of their article (what does this mean?). If published, this will include your full peer review and any attached files.

Reviewer #1: No

Reviewer #2: No

Reviewer #3: No

---

## [Editor Report · Acceptance letter]

10 Oct 2024

PONE-D-24-20348R1 

PLOS ONE

Dear Dr. Nebangwa, 

I'm pleased to inform you that your manuscript has been deemed suitable for publication in PLOS ONE. Congratulations! Your manuscript is now being handed over to our production team.

Kind regards, 

on behalf of

Dr. Anoop Kumar 

Academic Editor

PLOS ONE